# Construction of Two Independent RAB Family-Based Scoring Systems Based on Machine Learning Algorithms and Definition of RAB13 as a Novel Therapeutic Target for Hepatocellular Carcinoma

**DOI:** 10.3390/ijms24054335

**Published:** 2023-02-22

**Authors:** Chenhao Jiang, Zijian Liu, Jingsheng Yuan, Zhenru Wu, Lingxiang Kong, Jiayin Yang, Tao Lv

**Affiliations:** 1Department of Liver Transplantation Center, West China Hospital of Sichuan University, Chengdu 610041, China; 2Laboratory of Liver Transplantation, Frontiers Science Center for Disease-Related Molecular Network, West China Hospital of Sichuan University, Chengdu 610041, China; 3Department of Radiation Oncology, Cancer Center, West China Hospital of Sichuan University, Chengdu 610041, China; 4Laboratory of Pathology, Key Laboratory of Transplant Engineering and Immunology, NHC, West China Hospital of Sichuan University, Chengdu 610041, China

**Keywords:** hepatocellular carcinoma, Rab GTPase, tumor microenvironment, immune response, prognostic evaluation, risk model

## Abstract

Hepatocellular carcinoma (HCC) remains a global health challenge with a low early diagnosis rate and high mortality. The Rab GTPase (RAB) family plays an essential role in the occurrence and progression of HCC. Nonetheless, a comprehensive and systematic investigation of the RAB family has yet to be performed in HCC. We comprehensively assessed the expression landscape and prognostic significance of the RAB family in HCC and systematically correlated these RAB family genes with tumor microenvironment (TME) characteristics. Then, three RAB subtypes with distinct TME characteristics were determined. Using a machine learning algorithm, we further established a RAB score to quantify TME features and immune responses of individual tumors. Moreover, to better evaluate patient prognosis, we established a RAB risk score as an independent prognostic factor for patients with HCC. The risk models were validated in independent HCC cohorts and distinct HCC subgroups, and their complementary advantages guided clinical practice. Furthermore, we further confirmed that the knockdown of RAB13, a pivotal gene in risk models, suppressed HCC cell proliferation and metastasis by inhibiting the PI3K/AKT signaling pathway, CDK1/CDK4 expression, and epithelial-mesenchymal transition. In addition, RAB13 inhibited the activation of JAK2/STAT3 signaling and the expression of IRF1/IRF4. More importantly, we confirmed that RAB13 knockdown enhanced GPX4-dependent ferroptosis vulnerability, highlighting RAB13 as a potential therapeutic target. Overall, this work revealed that the RAB family played an integral role in forming HCC heterogeneity and complexity. RAB family-based integrative analysis contributed to enhancing our understanding of the TME and guided more effective immunotherapy and prognostic evaluation.

## 1. Introduction

Hepatocellular carcinoma (HCC) is the most common type of primary liver cancer and the fifth most common malignancy [1,2]. Moreover, HCC has been recognized as the primary cause of death in patients with liver cirrhosis [3]. Although many treatment options have been proposed in recent years, the prognosis of HCC patients is still unsatisfactory [4,5]. There is an urgent demand to discover early diagnostic markers and therapeutic targets, especially those that could be applied to modulate the tumor microenvironment (TME) and inhibit angiogenesis to improve the quality of life and prognosis of HCC patients. HCC is a morphologically heterogeneous malignancy with variable structural growth patterns and several distinct histological subtypes [6,7]. In recent years, large-scale attempts have been made to identify targeted genomic alterations in HCC [8,9]. However, translating genomic features into clinically personalized management remains a challenge for precision oncology.

The Rab GTPase (RAB) family is the most prominent in the Ras superfamily of small GTPases and comprises more than 60 members of humans [10]. Similar to other small GTPases, the RAB family is present intracellularly in the GTP-bound or GDP-bound form and regulates the transport of intracellular substances [11]. Some members of the RAB family are known to function in specific cells, where they control the trafficking of specialized vesicles [12]. Accumulating evidence has well-characterized the roles played by certain members of the RAB family in the progression of HCC. RAB40B and RAB11A promote HCC progression by regulating the PI3K/AKT signaling pathway and the expression of matrix metallopeptidase 2 (MMP2) [13,14]. You et al. reported that the hepatitis B virus X protein upregulates the oncogene RAB18, resulting in the dysregulation of lipogenesis and the proliferation of hepatoma cells [15]. Sui et al. emphasized that RAB31 promoted HCC progression by inhibiting cell apoptosis induced by the PI3K/AKT/Bcl-2/BAX pathway [16]. Nevertheless, their enormous number restricts the possibility of a comprehensive and thorough study of RAB family members. With the development of multiomics technologies, utilizing diverse gene expression profiles and bioinformatics approaches has provided the opportunity to define the expression patterns and clinical significance of RAB family members in HCC.

In this study, we characterized the expression landscapes of RAB family members across multiple datasets and summarized the biological characteristics of HCC with distinct expression patterns of the RAB family. Utilizing unsupervised clustering methods, RAB family-related molecular subtypes with distinct TME characteristics were determined based on a pooled HCC cohort. We further constructed a RAB score using the principal component analysis (PCA) score algorithm to predict the response to immunotherapy in HCC [17]. Moreover, to better guide the prognostic evaluation of patients, we constructed a RAB risk score using the least absolute shrinkage and selection operator (LASSO) Cox regression algorithm [18]. The predictive power of both risk models for the therapeutic efficacy of immune checkpoint inhibitors and the long-term prognosis were validated in independent HCC cohorts and distinct HCC subgroups. We further validated the role of RAB13 expression in cell proliferation and metastasis, and identified its potential downstream signaling pathways. Furthermore, we found that sorafenib could induce glutathione peroxidase 4 (GPX4)-dependent ferroptosis in RAB13-knockdown HCC, underscoring its potential as a therapeutic target for HCC.

## 2. Results

### 2.1. Expression, Diagnosis, and Prognosis of the RAB Family in HCC

The workflow of this study is depicted in Figure 1A. To exhibit expression alterations of the RAB family in HCC, we visualized the expression landscape of 64 RAB family members (Appendix A) available in The Cancer Genome Atlas (TCGA) and the International Cancer Genome Consortium (ICGC) cohorts. According to the criterion of *p* < 0.05, we observed that 42 RAB family genes in the TCGA cohort were markedly overexpressed, while five genes were significantly expressed at lower levels in HCC tissues than in paracancerous tissues (Figure 1B). However, in the ICGC cohort, 43 RAB family genes were highly expressed, and 8 genes were expressed at low levels in HCC tissues relative to paraneoplastic tissues (Appendix A). Moreover, the receiver operating characteristic (ROC) curve of the RAB family indicated that RAB24, RAB6B, RAB10, and RAB13 were excellent diagnostic predictors of HCC due to their area under the ROC curve (AUC) greater than 0.9. Meanwhile, the AUC values of 11 RAB members were greater than 0.8 (Figure 1C and Appendix A). These data indicated that RAB family members might be strictly associated with HCC initiation. Using univariate Cox regression analysis, we further investigated whether the expression of RAB family genes could predict the overall survival (OS) of HCC patients. As expected, in the TCGA cohort, we found that 26 RAB family genes were tightly associated with OS in HCC patients, and 2 of them (RAB10 and RAB29) were identified as “high-risk” factors for OS with a hazard ratio greater than 2 (Figure 1D). We further validated the critical roles of 23 RAB family genes in predicting OS in HCC using the ICGC dataset (Appendix A). Subsequently, we screened 15 critical genes of the RAB family based on the criteria of AUC values >0.7 and hazard ratio (HR) values of prognosis >1.0. Differential expression analysis revealed that all 15 genes were markedly overexpressed in HCC relative to paraneoplastic tissue (Figure 1E). Moreover, the expression correlations among the 15 RAB genes are shown in Figure 1F exhibiting a strong positive correlation with each other.

### 2.2. Biological Characteristics of Distinct RAB Clusters in HCC

The powerful hierarchical properties of the 15 RAB family genes in the diagnosis and prognosis prediction of HCC patients prompted us to further investigate their association with biological characteristics. First, we used the “Combat” algorithm to remove the batch effects of nontechnical bias between the hepatocellular liver carcinoma (LIHC) cohorts of TCGA and ICGC databases (Figure 2A) and named this combined gene expression profile the pooled HCC cohort to simplify subsequent analysis. Next, the nonnegative matrix factorization (NMF) algorithm was used to analyze the 15 RAB genes to characterize 2 RAB clusters in the pooled HCC cohort (Figure 2B). The silhouette width plots indicated that the silhouette width values of RAB cluster 1 and cluster 2 were 0.65 and 0.94 (Figure 2B), respectively, indicating good classification effectiveness of the NMF algorithm. Moreover, these 15 RAB family genes also differed markedly in distinct RAB clusters (Figure 2C,D). Kaplan–Meier survival analysis revealed that HCC patients in RAB cluster 1 had better OS than HCC patients in RAB cluster 2 (Figure 2B).

To investigate the biological characteristics of distinct RAB clusters, we further performed a Kyoto Encyclopedia of Genes and Genomes (KEGG) pathway enrichment analysis on the gene expression profiles of each RAB cluster in the pooled HCC cohort. Notably, both the gene ontology (GO)-biological processes and KEGG pathway enrichment analyses suggested that HCC in RAB cluster 1 was markedly associated with abnormal tumor metabolism, including metabolic pathways, drug metabolism, lipid metabolism, and amino acid metabolism, whereas HCC in RAB cluster 2 was significantly associated with aberrant activation of oncogenic signaling pathways, including PI3K/AKT signaling pathways, pathways in cancer, focal adhesion, and tight junction (Appendix A). Gene set enrichment analysis (GSEA) also revealed that metabolism-related signals and oncogenic signals were concentrated in RAB cluster 1 and RAB cluster 2, respectively, in HCC (Figure 2E). Interestingly, gene set variation analysis (GSVA) indicated that HCC of RAB cluster 1 not only had the most remarkable correlation with metabolic pathways such as xenobiotic metabolism, bile acid metabolism, and fatty acid metabolism, but also had a strong association with immune signals such as the interferon-gamma/alpha response, while RAB cluster 2 in HCC was significantly associated with cell cycle regulation, the TGF-β signaling pathway, the PI3K/AKT/mTOR signaling pathway, and epithelial-mesenchymal transition (EMT) (Appendix A).

Significant progress has been achieved recently using immunotherapy for the treatment of HCC [19,20], highlighting the importance of the immune microenvironment in the treatment of HCC. According to previously reported algorithms [17], we further analyzed the differences in the proportion of immune cell infiltration between RAB clusters in the pooled HCC cohort. Interestingly, the analysis indicated that the HCC of RAB cluster 1 had higher levels of B cells, CD8 T cells, dendritic cells (DCs), activated DCs (aDCs), cytotoxic cells, eosinophils, and neutrophils, whereas RAB cluster 2 in HCC contained a higher proportion of macrophages, mast cells, natural killer (NK) cells, and T helper cells (Figure 2F). We further performed a tumor immune dysfunction and exclusion (TIDE) analysis on the pooled HCC cohort to evaluate the association of RAB clusters with immunotherapy response in HCC. As expected, RAB cluster 1 had a lower TIDE score and a better response to immunotherapy than RAB cluster 2 in HCC (Figure 2G,H). Moreover, our TIDE analysis revealed that RAB cluster 1 in HCC had lower levels of cancer-associated fibroblasts (CAFs) and myeloid-derived suppressor cells (MDSCs), as well as a higher microsatellite steady state (MSI), whereas HCC of RAB cluster 2 was directly associated with immune exclusion (Figure 2G). However, the levels of PD-L1, CD8, interferon-gamma, and CD8 and T-cell inflammation (Merck18) were not markedly different between the two RAB clusters in HCC (Figure 2G). The above results suggested that RAB family genes may play critical roles in the progression and TME cell infiltration of HCC.

### 2.3. Three TME Subtypes Were Revealed by Unsupervised Clustering Analysis of the RAB-Associated Signatures in HCC

The absence of differences in immune checkpoints and partial immune cells indicated that the 15 RAB family genes alone failed to cluster HCC well (Figure 2G). Thus, we further attempted to perform unsupervised clustering for differentially expressed genes (DEGs) in the two RAB clusters to identify the subtypes of HCC. According to the criteria of log | fold change (FC)| > 1 and *p* < 0.05, a total of 830 DEGs between the 2 RAB clusters from the pooled HCC cohort were obtained, which were named RAB-associated gene signatures, including 304 positively correlated genes and 526 negatively correlated genes (Appendix A). Subsequently, the patients with different TME patterns in the pooled HCC cohort were classified based on the expression of RAB-associated gene signatures using the R package ConsensusClusterPlus. Notably, three distinct RAB subtypes were eventually identified using unsupervised clustering, including 326 cases in subtype-1, 69 cases in subtype-2, and 213 cases in subtype-3 (Figure 3A). Prognostic analysis for the three main RAB subtypes revealed a particularly prominent survival advantage in subtype-1 (Figure 3B).

To investigate the biological features of these distinct RAB subtypes, we performed a GO enrichment analysis. As shown in Figure 3C, the results of the GO biological process analysis implied that RAB subtype-1 was markedly enriched in metabolic pathways, RAB subtype-2 presented enrichment pathways associated with carcinogenic activation, and RAB subtype-3 was prominently associated with immune activation. GSVA further indicated that adipogenesis, cholesterol homeostasis, fatty acid metabolism, and bile acid metabolism were markedly activated in RAB subtype-1 but were remarkably inhibited in RAB subtype-2. Additionally, immune response-related signals such as the interferon-gamma/alpha response, IL6/JAK/STAT3 signaling pathway, and IL2/STAT5 signaling pathway were significantly activated in RAB subtype-3 (Figure 3D). For further quantitative comparison, we performed GSVA for metabolic-, immune-, and carcinogenesis-related signaling pathways. Consistently, glycolysis, heme metabolism, adipogenesis, and fatty acid metabolism had the highest enrichment scores (ES) in RAB subtype-1 compared with RAB subtype-2 or -3, while RAB subtype-3 had the highest ES in the immune response-related signaling pathways (Figure 3E). Meanwhile, RAB subtype-2 had a higher ES in oncogenic-related signaling, including cell cycle-related signaling, the PI3K/AKT/mTOR signaling pathway, Notch signaling, and the P53 pathway (Figure 3F).

Interestingly, the analysis of TME cell infiltration revealed that the three RAB subtypes had distinct immune cell infiltration characteristics. Specifically, RAB subtype-3 had the highest abundance of adaptive immune cells, including B cells, T cells, NK cells, and neutrophils (Figure 3G). Together with the activation status of its immune signaling, RAB subtype-3 was classified as an immune-inflamed phenotype, a previously reported model of immune classification characterized by immune activation and adaptive immune cell infiltration [21]. However, RAB subtype-2 was remarkably abundant in innate immune cell infiltration, including aDCs, immature DCs (iDCs), macrophages, mast cells, T helper cells, central memory T cells (Tcm), effector memory T cells (Tem), follicular helper T cells (Tfh), and Th1 cells (Figure 3G). However, patients with RAB subtype-2 HCC did not exhibit a matching survival advantage (Figure 3B). Previous studies have demonstrated that stromal activation suppresses the antitumor effects of immune cells [21]. GSEA analysis revealed that stromal activity was markedly enhanced in RAB subtype-2, including the activation of apical surfaces and junctions, TGF-β signaling pathways, and EMT (Figure 3F). Thus, RAB subtype-2 was classified as an immune-excluded phenotype characterized by stromal activation and innate immune cell infiltration. Notably, malignant tumors with immune exclusion also exhibited the presence of abundant immune cells, but these cells remained in the stroma surrounding tumor cell nests rather than penetrating the parenchyma and were considered T-cell suppressive. Furthermore, RAB subtype-1 in HCC had only plasmacytoid DC, eosinophil, and T regulatory cell (Treg) infiltration (Figure 3G), which was classified as an immune-desert phenotype characterized by the suppression of immunity.

We further found that the HCC of RAB subtype-2 had the highest mRNAsi index compared with RAB subtype-1 and -3, representing the strongest tumor stemness in RAB subtype-2 in HCC (Figure 3H). Conversely, RAB subtype-2 had the lowest ferroptosis index, indicating that this subtype of HCC had the weakest ferroptosis vulnerability, while subtype-1 had the highest ferroptosis index (Figure 3I). Moreover, a tumor mutational burden (TMB) analysis revealed that the overall TMB was significantly higher in RAB subtype-1 than in the other subtypes (Figure 3J), with mutations mainly originating from catenin beta 1 (CTNNB1), whereas the mutations in RAB subtype-2 and -3 were primarily derived from P53 (Figure 3K). Based on the above analysis, we realized that HCC could be classified into three subtypes with distinct TME characterization based on RAB-associated gene signatures, namely: RAB subtype-1, oncogenic signal suppression, metabolic activation, immune-desert, tumor stemness, ferroptosis sensitivity, and high TMB; RAB subtype-2, oncogenic signal activation, metabolic suppression, immune-excluded and ferroptosis tolerance; and RAB subtype-3, oncogenic signal suppression, metabolic suppression, and immune-inflamed.

### 2.4. Construction of a RAB Score and Evaluation of Its Predictive Ability in the Pooled HCC Cohort

The above results indicated that RAB-associated gene signatures played a nonnegligible role in shaping distinct TME landscapes in HCC. Next, we further evaluated whether RAB-associated gene signatures could predict TME characteristics and prognosis in individual patients. According to the criteria of log |FC| > 2 and *p* < 0.05, we further screened 100 DEGs between 2 RAB subtypes from the pooled HCC cohort to narrow the gene number of RAB-associated gene signatures and facilitate subsequent analysis (Figure 4A). Then, based on these 100 phenotype-related DEGs, we constructed a scoring system to quantify the TME characteristics of individual patients with HCC, which was termed the RAB score. Notably, the Kruskal–Wallis test showed considerable differences in RAB scores between RAB subtypes (Figure 4B). RAB subtype-1 exhibited the highest median score, while RAB subtype-2 had the lowest median score, which implied that a high RAB score could be closely associated with metabolic activation-related signatures, whereas a low RAB score could be related to oncogenic signal activation-related signatures. Consistently, as shown in Figure 4C, GSVA revealed that metabolism-related signals were markedly activated in HCC patients with high RAB scores, including xenobiotic metabolism, bile acid metabolism, and fatty acid metabolism, while oncogenic-related signaling pathways were remarkably activated in HCC patients with low RAB scores, including cell cycle-related signaling, EMT, the PI3K/AKT/mTOR signaling pathway, and the TGF-β signaling pathway. Interestingly, the immune-related interferon alpha response was positively correlated with the RAB score, while the IL2/STAT5 signaling pathway was negatively correlated with the RAB score.

Next, patients in the pooled HCC cohort were divided into low or high RAB score groups with a median as the cutoff value. Notably, patients with high RAB scores demonstrated a moderate survival benefit (*p* < 0.001, 95% CI: 0.36–0.66), with an HR value of 0.49 (Figure 4D). Moreover, the mRNAsi index confirmed that a high RAB score HCC was markedly correlated with lower tumor stemness (Figure 4E). Meanwhile, the RAB score and ferroptosis index also exhibited a noticeable positive correlation (Figure 4F). Subsequently, the analyses of TME cell infiltration indicated that HCC with a higher RAB score was correlated significantly with a high proportion of Th17 cell, B cell, DC cell, eosinophil, and neutrophil infiltration (Figure 4G), which meant that these patients were characterized by an immune-inflamed phenotype with a better clinical outcome. However, HCC patients with a lower RAB score were strongly correlated with the proportion of NK cells, Tem cells, Tfh cells, T helper cells, macrophages, and mast cell infiltration (Figure 4G), which also indicated that these HCC patients tend to have an immune-excluded phenotype with a poorer clinical outcome. TIDE analysis was further performed in the pooled HCC cohort to evaluate the capability of the RAB score in predicting the response to immunotherapy. Notably, studies confirmed that high TIDE scores were associated with poorer responses to anti-PD1 and anti-Cytotoxic T-lymphocyte-associated protein 4 (CTLA-4) therapy [22]. Our TIDE quantitative analysis indicated that HCCs with high RAB scores usually had lower TIDE scores, suggesting that HCC patients with high RAB scores responded better to immunotherapy (Figure 4H). Moreover, a low RAB score was directly associated with immune exclusion, which further emphasized that HCC with a low RAB score was closely related to the immune exclusion phenotype. Meanwhile, the correlation analysis further indicated that HCC with a low RAB score contained more abundant immunosuppressive cells, including MDSCs and CAFs (Figure 4H). These data all support the essential indicative and predictive role of the RAB score in the TME of HCC.

### 2.5. Validation of the RAB Score in Response to Immunotherapy

To further validate the validity of the RAB score for predicting immunotherapy response, multiple HCC datasets from the Gene Expression Omnibus (GEO) database were used as test cohorts to validate our above results. Pathway enrichment analysis of GSE14520 indicated that the RAB score was positively correlated with metabolism-related signals but adversely correlated with oncogenic signaling pathways such as the cell cycle, EMT, and inflammatory responses (Figure 5A, upper left panel). Moreover, the RAB score was significantly positively correlated with the proportion of Th17 cell, DC cell, eosinophil, and neutrophil infiltration but negatively associated with NK cell, Tfh cell, T helper cell, macrophage, and mast cell infiltration levels in HCC (Figure 5A, lower right panel). The analysis of major histocompatibility complex (MHC) molecular and adhesion molecule levels further indicated that HCC with a high RAB score has higher levels of PDCD1, CD40, and ICAM4 factors (Figure 5B), suggesting that this type of HCC may have better immune responses than HCC with a low RAB score. More importantly, a low RAB score demonstrated significant clinical benefit and significantly prolonged OS and recurrence-free survival (RFS) compared with a high RAB score in HCC (Figure 5C). Furthermore, consistent with the TIDE analysis of the pooled HCC cohort, the TIDE analysis of GSE14520 also suggested that a high RAB score was indeed associated with low TIDE scores and a higher proportion of patients with immune responses in HCC (Figure 5D), and these results were also consistently validated by TIDE analysis of GSE5975, GSE25097, and GSE124751 (Figure 5E–G).

### 2.6. Construction of the RAB Risk Score to Better Predict the Prognosis of HCC Patients

Notably, the previous prognostic analysis showed that the RAB score could well predict the prognosis of HCC patients with a *p* value significantly less than 0.05 (Figure 4D and Figure 5C). However, the HR values of the prognostic analysis were all less than 0.6, which indicated that the prognostic stratification ability of the RAB score was not very reliable. To further obtain an ideal prognostic prediction model based on RAB-associated gene signatures (Appendix A), we applied an iterative LASSO Cox regression algorithm. Interestingly, we obtained 26 genes with independent prognostic significance in patients with HCC (Figure 6A). Then, the RAB risk score was calculated based on the expression values and regression coefficients of these 26 genes (Figure 6B and Appendix A). Importantly, using the pooled HCC cohort, we found that these 26 genes could well predict the prognosis of HCC patients with a 5-year AUC value of 0.765 and an HR value of 3.78 (Figure 6C,D). The RAB risk score allowed patients in the pooled HCC cohort to be divided into high-risk (n = 304, score value > 1.013) and low-risk (n = 304, score value < 1.013) score groups based on median values. Consistently, the number of deaths in HCC patients increased significantly with increasing RAB risk score (Figure 6E), which also reflected that the high-risk score group had a significantly higher mortality rate than the low-risk score group. Subsequently, we attempted to determine whether the RAB risk score could serve as an independent prognostic factor in HCC patients by univariate and multivariate Cox regression analyses. As expected, the univariate analysis demonstrated that tumor stage, tissue grade, RAB score, and RAB risk score were all prognostic factors for HCC patients (Figure 6F). Multivariate Cox regression analysis further indicated that RAB risk score and tumor stage were independent factors that could be used to predict the prognosis of HCC patients (Figure 6G). To provide clinicians with a relatively quantitative tool for predicting mortality risk in HCC patients, we constructed a nomogram using these prognostic factors (Figure 6H). By adding the points for each prognostic factor, each patient was assigned a total prognostic score. A higher total prognostic score corresponds to a worse OS outcome in patients with HCC. The calibration curves suggested good consistency between the prediction by the nomogram and actual OS outcomes at three and five years (Figure 6I). More importantly, the time-dependent AUC values of the RAB risk score for predicting the 1- to 8-year survival rates were all greater than 0.75, which was much better than the time-dependent AUC value of the RAB score for predicting OS (Figure 6J).

### 2.7. Validation of the Prognostic Predictive Ability of the RAB Risk Score

To determine whether the RAB risk score is robust, we further evaluated the predictive effect of the RAB risk score on the prognosis of HCC patients in different clinical cohorts and subgroups. Here, the median value was used as the cutoff value for different HCC cohorts. First, we validated the prognostic stratification ability of the RAB risk score using the TCGA and ICGC cohorts. Kaplan–Meier survival curves of the TCGA cohort indicated that HCC patients with high RAB risk scores had worse OS than those with low RAB risk scores (*p* < 0.001, HR = 3.25, 95% CI = 2.27–4.65), with a time-dependent AUC value greater than 0.75 at 1, 3, 5, and 8 years (Figure 7A). Consistent Kaplan–Meier analysis outcomes were obtained from the ICGC cohort (*p* < 0.001, HR = 5.13, 95% CI = 1.76–9.56). The time-dependent AUC values of the RAB risk score for the prediction of one- to four-year survival rates in the ICGC cohort all exceeded 0.8 (Figure 7B). Subsequently, all patients in the pooled HCC cohort were grouped by age and then ranked by the RAB risk score into high- and low-risk subgroups. Kaplan–Meier survival analyses indicated that the OS in the high-risk subgroup was markedly worse than that in the low-risk subgroup (age >60: *p* < 0.001, HR = 4.34, 95% CI = 2.95–6.39; age ≤60: *p* < 0.001, HR = 3.11, 95% CI = 1.91–5.06) (Figure 7C). Our above analysis suggested that tumor stage, tissue grade, and RAB score were prognostic factors in patients with HCC. Likewise, a high RAB risk score was correlated with dramatically worse OS regardless of whether the patient exhibited early- (*p* < 0.001, HR = 3.52, 95% CI = 2.27–5.47) or advanced-stage (*p* < 0.001, HR = 3.51, 95% CI = 2.11–5.81), well-differentiated (*p* < 0.001, HR = 3.34, 95% CI = 2.20–5.07) or poorly differentiated (*p* < 0.001, HR = 5.91, 95% CI = 3.32–10.53), and high- (*p* < 0.001, HR = 4.03, 95% CI = 2.63–6.17) or low-RAB score (*p* < 0.001, HR = 3.81, 95% CI = 2.06–7.04) HCC (Figure 7D-F). Tumor mutation is also a malignant burden factor of HCC. Consistently, the RAB risk score provided a statistical stratification of OS regardless of whether the HCC was CTNNB1 wild-type (WT) (*p* < 0.001, HR = 3.14, 95% CI = 2.06–4.80), CTNNB1 mutant (MUT) (*p* < 0.001, HR = 3.48, 95% CI = 1.64–7.38), P53 WT (*p* < 0.001, HR = 3.09, 95% CI = 2.01–4.74), or P53 MUT (*p* < 0.001, HR = 5.56, 95% CI = 2.87–10.78) (Figure 7G,H). Furthermore, we further validated the predictive power of the RAB risk score for OS and RFS in HCC patients using the GSE14520 dataset. As expected, Kaplan–Meier survival curves of the GSE14520 dataset also indicated that HCC patients with high risk scores had worse OS (*p* < 0.001, HR = 2.44, 95% CI = 1.59–3.75) and RFS (*p* < 0.001, HR = 1.98, 95% CI = 1.38–2.86) than those with low risk scores with a time-dependent AUC value greater than 0.60 at 1, 3, and 5 years (Figure 7I,J). These data demonstrated that the RAB risk score is a reliable and stable model for predicting the prognosis of patients with HCC.

### 2.8. RAB13 Is Essential for the Malignant Biological Behaviors of HCC Cells

The above data proposed and validated a RAB score for immune response prediction and a RAB risk score for prognosis prediction in HCC. Here, Pearson correlation analysis further indicated an inverse correlation between the RAB score and the RAB risk score (Figure 8A). Next, we attempted to further screen critical RAB family genes to ascertain their roles in HCC. We performed TIDE analysis on 15 previously screened RAB family members (Figure 1E), and the results indicated that the expression of seven RAB family genes (RAB11A, RAB13, RAB1B, RAB35, RAB5B, RAB5C, and RAB6B) had remarkable differences in the immune response of HCC (Figure 8B). Notably, RAB13 was used as a target for subsequent studies, as its roles were not fully explored in HCC. Further analysis revealed that RAB13 exhibited a positive correlation with the RAB score and a negative correlation with the RAB risk score (Figure 8C,D). GSVA showed that RAB13 was positively correlated with multiple oncogenic signaling pathways, including the PI3K/AKT signaling pathway, EMT, and cell cycle-related signaling pathways, while it was negatively associated with metabolism-related signaling pathways. Moreover, RAB13 also exhibited a marked negative correlation with immune-related signals, including the IL2/STAT5 signaling pathway, IL6/STAT3 signaling pathway, inflammatory response, and interferon alpha/gamma response (Figure 8E). We further analyzed the correlation between RAB13 and the immune microenvironment. Interestingly, the expression of RAB13 was markedly positively correlated with immune exclusion (Figure 8F). Meanwhile, RAB13 expression was positively associated with MDSC, TAM M2, and Th2 cell levels and remarkably negatively correlated with neutrophil, eosinophil, DC cell, cytotoxic cell, CD8 T cell, and B-cell infiltration levels in HCC (Figure 8F,G). In addition, the level of RAB13 was negatively correlated with the immune checkpoint PDL1 (Figure 8F).

Furthermore, our clinical samples indicated that RAB13 protein expression was markedly elevated in HCC tissues compared to paired non-cancer liver (NCL) tissues (Figure 9A). Using the Human Protein Atlas, we also verified that the protein expression of RAB13 was markedly higher in HCC than in paracancerous tissues (Figure 9B). Next, we further investigated the potential role of RAB13 in HCC using cytological assays. RAB13 was markedly knocked down by transfection with siRNA targeting RAB13 sequences (siRAB13) in Huh7 and Hep3B cells compared to the control siRNA (siCTL) (Figure 9C,D). The Cell Counting Kit-8 (CCK-8) and EdU assays revealed that RAB13 knockdown markedly inhibited the proliferation and DNA replication of HCC cells (Figure 9E,F). Moreover, wound healing and transwell assays demonstrated that RAB13 silencing significantly inhibited the metastasis of HCC cells (Figure 9G,H). Based on previous analysis, we further investigated the precise relationship of RAB13 expression with the PI3K/AKT signaling pathway, cell cycle regulation, and EMT. As expected, RAB13 silencing markedly suppressed the protein levels and phosphorylation levels of the PI3K/AKT signaling pathway (Figure 10A). CDK1 is pivotal in regulating the G2-phase transition of the cell cycle, while CDK4 manages the G1 phase to enter the S phase of DNA synthesis [23,24]. Interestingly, our results indicated that RAB13 knockdown significantly restrained CDK1 and CDK4 expression (Figure 10B). In addition, inhibition of RAB13 expression restricted the EMT process (Figure 10C). Notably, our data implied that RAB13 levels are inversely correlated with the IL2/STAT3 signaling pathway (Figure 8E). Consistently, RAB13 knockdown promoted the expression and activation of JAK2/STAT3 signaling (Figure 10D). Moreover, we found that RAB13 silencing enhanced the expression of interferon regulatory factors-1 (IRF1) and IRF4 (Figure 10E), which are vital factors mediating tumor immunity. These data indicate that elevated RAB13 expression is critical for the malignant progression of HCC.

### 2.9. RAB13 Knockdown Promotes GPX4-Dependent Ferroptosis Vulnerability in HCC Cells

Our data suggested that activation of metabolism-related signaling pathways was associated with a better prognosis for patients with HCC (Figure 3B,E). Notably, dysregulation of metabolic signaling regulates ferroptosis vulnerability. Therefore, we wondered whether RAB13 expression could alter the ferroptosis vulnerability of HCC cells. Correlation analysis indicated a significant negative relationship between RAB13 expression and the FPI index (Figure 10F), implying that RAB13 overexpression may impair ferroptosis vulnerability in HCC. In addition, sorafenib, a ferroptosis inducer, markedly restrained the proliferation of RAB13-knockdown HCC cells (Figure 10G). To further demonstrate that RAB13-inhibited HCC cells suffered ferroptosis following sorafenib treatment, we observed the Phen Green SK diacetate (P-GSK) probe and examined the variation in malondialdehyde (MDA) levels. As expected, the P-GSK probe indicated that sorafenib markedly promoted the accumulation of iron in HCC cells after RAB13 knockdown (Figure 10H). Meanwhile, sorafenib promoted lipid oxidative damage in RAB13-knockdown HCC cells (Figure 10I). Finally, we detected alterations in GPX4 and ferroptosis suppressor protein 1 (FSP1) expression. Interestingly, our Western blot results indicated that RAB13 knockdown suppressed GPX4 protein expression but not FSP1 expression (Figure 10J). These data illustrate that RAB13 is a crucial target for boosting GPX4-dependent ferroptosis vulnerability in HCC.

## 3. Discussion

The RAB family acts as molecular switches that localize to different intracellular membranes, providing spatiotemporal control of organelle maintenance and trafficking [10,11,12]. However, a comprehensive and thorough investigation of RAB family genes in HCC is still lacking. Here, we comprehensively characterized the landscape of RAB family genes and constructed RAB gene-related models for the clustering and evaluation of HCC, which has tremendous clinical implications.

In our study, we first attempted to cluster the gene expression profiles of HCC according to RAB family genes using the NMF algorithm. Although the results suggested that RAB cluster 1 and RAB cluster 2 could well stratify the prognosis of HCC patients, the two clusters were more similar to a summary of the biological characteristics for the gene expression patterns with different levels of RAB family genes and failed to exhibit good stratification in describing TME differences. Furthermore, the mRNA transcriptome differences between distinct RAB expression levels have been demonstrated to be dramatically associated with metabolic-, oncogenic-, and immune-related biological pathways. Thus, these DEGs were considered RAB-associated signatures. Interestingly, three genomic subtypes with distinct TME patterns were revealed based on RAB-associated signatures utilizing unsupervised clustering analysis. RAB subtype-1 was characterized by the activation of metabolism and the suppression of oncogenic signaling corresponding to the immune-desert phenotype. In addition, RAB subtype-1 had the highest TMB and the weakest ferroptosis vulnerability. Notably, RAB subtype-1 exhibited greater prognostic survival than RAB subtypes-2 and -3, indicating that immune status is not an independent predictor for assessing patient prognosis. Moreover, we hypothesize that the worst prognosis in RAB subtype-2 is associated with its oncogenic signaling activation, ferroptosis tolerance, and immune-excluded phenotype. Furthermore, our analysis revealed that patients with RAB subtype-2 are optimal candidates for immune checkpoint therapy, as RAB subtype-2 was characterized by activation of adaptive immunity, corresponding to an immune-inflamed phenotype, also known as an immune hot tumor [25,26], manifested by a prominent infiltration of immune cells in the TME.

Considering individual heterogeneity, to further quantify individual tumor characteristics and facilitate clinical application, we attempted to establish a scoring system—the RAB score—to evaluate the immunological features and prognosis of individual HCC patients. RAB subtype-1, characterized by an immune-desert phenotype, exhibited a higher RAB score and was associated with a better prognosis. RAB subtype-2, which is characterized by an immune-excluded phenotype, showed a lower RAB score and was associated with a poorer prognosis. Moreover, we validated this model in several distinct HCC cohorts. This finding indicated that the RAB score was a robust and reliable tool to comprehensively assess the TME characteristics for individual HCC, which could be used to further determine the tumor immunophenotype. However, the integrated analysis revealed that the RAB score was not an independent prognostic biomarker for HCC. To this end, we further constructed the RAB risk score using the iterative LASSO regression algorithm, whose predictive power of prognosis was also validated in various HCC cohorts and subgroups. Notably, the RAB risk score was not comparable to the RAB score in evaluating the TME characteristics of HCC, so we did not present and interpret these results. We speculated that this status was mainly caused by the fact that the RAB score recombined the crucial genes of the RAB family-related DEGs through the PCA score method, which contains multiple gene patterns and could well characterize the TME features according to the expression of crucial genes, but not all of these essential genes were prognostic stratification genes; thus, the RAB score was not suitable for prognostic assessment of HCC. Conversely, the RAB risk score incorporated genes with significant prognostic stratification, but its limited number of genes restricts the capacity of a robust depiction of TME for HCC. Therefore, the two scoring models could complementarily guide clinicians in the management of HCC, which has potential clinical significance.

The role of RAB13 in tumors has been widely reported. Wang et al. elucidated that RAB13 sustains breast cancer stem cells by supporting tumor-stromal crosstalk [27]. Hinger et al. reported that RAB13 regulates the secretion of small extracellular vesicles in mutant KRAS colorectal cancer cells [28]. Zhang et al. demonstrated that RNF115 inhibits the postendoplasmic reticulum trafficking of Toll-like receptors (TLRs) and TLR-mediated immune responses by catalyzing K11-linked RAB1A and RAB13 ubiquitination [29]. However, the role of RAB13 in HCC has not been reported. In our constructed models, we found that RAB13 had significant weights in both the RAB score predicting immune response and the RAB risk score predicting prognosis. Therefore, we further investigated the function of RAB13 in HCC cells using cytological studies. Interestingly, we found that RAB13 could be involved in modulating HCC cell proliferation through the PI3K/AKT signaling pathways and cell cycle regulation. Meanwhile, we demonstrated that RAB13 promotes the metastasis of HCC cells through EMT. Moreover, we found that the promotion of the immune-excluded phenotype by elevated RAB13 expression may be associated with the inhibition of interferon-regulated signaling (IRF1/IRF4) and the JAK2/STAT3 signaling pathway. These data all indicated that RAB13 might be a potential target for HCC therapy. To validate this hypothesis, we further tested whether RAB13 could regulate ferroptosis due to the relevance of RAB13 to metabolism-related signaling and ferroptosis vulnerability. As expected, RAB13-silenced HCC had increased sensitivity to sorafenib, and this phenomenon was associated with the accumulation of intracellular iron and increased levels of lipid oxidation. More importantly, our Western blotting results confirmed that RAB13-induced alterations in ferroptosis vulnerability were dependent on GPX4 expression.

## 4. Materials and Methods

### 4.1. Clinical Samples and Immunohistochemistry

A retrospective analysis of resected HCC samples at West China Hospital of Sichuan University from May 2014 to December 2020 was performed. Thirty fresh human HCC and paired NCL tissues were collected. Immunohistochemistry (IHC) staining was performed as described previously [30,31]. Anti-RAB13 (ABclonal, A10571, Wuhan, China, 1:200) was used. The IHC results were evaluated by two independent observers based on the percentage of positively stained cells (scored from 0 to 3 points) and intensity of staining (scored from 0 to 3 points), and a final immunoreactivity score (range 0–9 points) was obtained by multiplying the two scores. RAB13 expression levels were classified as low if the score was less than five and high if the score was ≥ five [30,31].

This study was approved by the Ethics Committee on Biomedical Research, West China Hospital of Sichuan University (2020, No 385). Informed consent forms were signed by all involved patients or their families.

### 4.2. Cell Culture and Reagents

Huh7 and Hep3B cell lines were purchased from the National Collection of Authenticated Cell Cultures (Shanghai, China) and were cultured in complete medium containing Dulbecco’s modified Eagle’s medium (HyClone, Logan, UT, USA) supplemented with 10% fetal bovine serum (Gibco, Grand Island, NY, USA), 1000 U/mL penicillin, and 100 μg/mL streptomycin (HyClone, Logan, UT, USA), and were grown in a humidified air atmosphere containing 5% CO_2_ at 37 °C. All cell lines were analyzed by STR profiling for cell line authentication and routine mycoplasma detection. Sorafenib (S7397) was purchased from Selleckchem (Houston, TX, USA).

### 4.3. Transfection

Transfection was performed as previously described [30,31]. Additional information about siRNAs is available in Appendix A.

### 4.4. Quantitative Real-Time Polymerase Chain Reaction (qRT–PCR) and Western Blot Analysis

qRT–PCR and Western blot analysis were performed as previously described [30,31]. The primers and the primary antibodies used in this study are listed in Appendix A, respectively.

### 4.5. Wound Healing and Transwell Assays

Wound healing assays were performed as previously described [31]. For the transwell assay, transfected HCC cells resuspended in an FBS-free medium were added to the top chamber (Corning-Costar; pore size 8 μm), and the bottom chamber was filled with 30% FBS as an inducer. After 48 h, the cells that failed to invade from the top of the membranes were erased, and then the invaded cells on the bottom of the membrane were fixed and stained. Invaded cells from five random fields were counted and photographed under a light microscope.

### 4.6. Cell Counting Kit-8

CCK-8 proliferation assay was performed as previously described [30]. Additionally, to examine the inhibitory effect of sorafenib on the indicated cells, the processed cells (1 × 10^3^ cells per well) were inoculated in 96-well plates for 24 h. Sorafenib was then administered at a concentration of 5 μM and incubated for 72 h. Then, 10 μL of CCK-8 solution was added to the wells and incubated for 4 h. Finally, the absorbance at 450 nmol was recorded, and the results were analyzed.

### 4.7. EdU Assays

EdU assays were performed using a BeyoClick™ EdU Cell Proliferation Kit with Alexa Fluor 594 (Beyotime, Wuhan, China) according to the manufacturer’s instructions.

### 4.8. Ferroptosis Detection

A P-GSK probe was used to monitor the iron content in the indicated HCC cells using a Phen Green SK Reagent Kit (Thermo, Waltham, MA, USA) in accordance with the manufacturer’s instructions. The levels of MDA (A003-1-2) were measured to assess the level of lipid oxidative damage using commercially available kits from Nanjing Jiancheng Bioengineering Institute (Nanjing, China) in accordance with the manufacturer’s instructions.

### 4.9. Data Sources and Preprocessing

The LIHC clinical information and raw fragment per kilobase (FPKM) values were taken from the ICGC and TCGA datasets. We then transformed FPKM values into transcripts per kilobase million (TPM) values. The series matrix files of the Affymetrix and Illumina-generated microarray for GSE14520, GSE5975, GSE25097, and GSE124751 were directly downloaded from the GEO database.

### 4.10. Pathway Enrichment Analysis

The Database for Annotation, Visualization, and Integrated Discovery (DAVID) v6.8 was used for the KEGG pathway analysis. According to previously published expression methods, we further performed the GSEA on the specified set of transcripts. Moreover, the gene set “c5.all.v6.2. symbols” was downloaded from the MSigDB database, and another published pathway gene set is summarized in Appendix A. GSVA enrichment analysis was further used to estimate the pathway and biological process variations using these two gene sets.

### 4.11. Consensus Clustering with Nonnegative Matrix Factorization

To correlate the survival status of patients with gene expression values, we employed a consensus clustering method, NMF, to perform clustering analysis based on the expression of RAB genes and the OS of HCC patients [32]. The principle of consensus clustering is to perform two-dimensional resampling of the original dataset and then repeatedly cluster the perturbation subsets, and the final clustering results are obtained by clustering the consensus matrix. Pearson’s correlation coefficient was used to measure the distance, and “average” was used as the linkage method, with 100 repetitions. The performance of these clustering methods was evaluated with three frequently utilized measures as previously reported [33]: (1) Survival analysis to evaluate the prognostic values between subtypes; (2) average silhouette width, a measure of cluster coherence, to assess the similarities across subtypes; and (3) clustering heatmap to intuitively visualize the effect of sample clustering.

### 4.12. Unsupervised Clustering for RAB-Associated Gene Signatures

Based on the expression of RAB-associated gene signatures, unsupervised clustering analysis was conducted using the pooled HCC cohort to identify distinct HCC subtypes for further research. The number of clusters and their stability were determined by a consensus clustering algorithm. The above steps were repeated 1000 times using the “ConsensuClusterPlus” R package to ensure the strength of the classification.

### 4.13. Construction of the Risk Models

To quantify the RAB expression patterns of individual tumors, the PCA score method was used to construct a scoring system named the RAB score [17]. In addition, the iteration LASSO Cox regression model was used to screen for the best genes for prognostic assessment in HCC [18]. The RAB risk score could be calculated using the following formula: RAB risk score = Σ (Coef i × Exp i), where I is the member involved in the gene signature.

### 4.14. Immune Response Prediction and Immune Microenvironment Assessment

The TIDE algorithm was used to predict HCC responsiveness to immunotherapy [23]. We used the GSVA method to quantify the relative abundance of each infiltrating cell in a single sample. The immune cell markers used in this study were extracted from a previously published authoritative study.

### 4.15. mRNA-Based Stemness Index (mRNAsi) and Ferroptosis Potential Index (FPI)

To assess the stemness of cancer cells, a one-class logistic regression algorithm, mRNAsi, was used to calculate the stemness index for each HCC sample using the workflow available on a previously established database [34]. In addition, an index representing ferroptosis vulnerability was found from the expression data of ferroptosis core machine genes according to a previously published algorithm [35].

### 4.16. Development and Validation of the Prognostic Nomogram

Based on the clinical risk factors and multivariate Cox regression coefficients, a prognostic nomogram was built using the “rms” R package, and the predictive accuracy of this nomogram was assessed using the calibration curve and the concordance index.

### 4.17. Statistical Analysis

All statistical analyses were performed using R software (version 3.6.1). Analysis of differentially expressed genes (DEGs) between different defined groups was performed using the “limma” R package. DEGs between the two RAB clusters were obtained with significance criteria set as adjusted *p* value < 0.05 and log2 |FC| > 1, while the criteria of *p* value < 0.05 and log2 |FC| > 2 were set for screening DEGs between two RAB subtypes. DEGs were visualized as heatmaps in R using the packages “pheatmap” and “ggplot2”. To calculate the TMB per megabase, the total number of mutations counted was divided by the size of the coding region of the targeted territory in the TCGA-LIHC cohort. The mutation landscape oncoprint was generated using the R package “ComplexHeatmap”. The comparison of normally distributed variables between the two groups was performed using an unpaired t-test, and the statistical significance of the nonnormally distributed variables was estimated using the Mann–Whitney U test (Wilcoxon rank-sum test). Spearman’s correlation analysis was performed to calculate the correlation coefficient between the two factors. Based on the correlation between gene expression and patient survival, the optimal cutoff point for each dataset was determined using the “survminer” R package, and the “surv-cutpoint” function was used to repeat all potential cutoff points to obtain the maximum rank statistic, divided into two groups: high and low. Survival curves for prognostic analysis were generated using the Kaplan–Meier method, and significant differences were determined using the log-rank test. The false discovery rate (FDR) method was used to adjust the *p* value for multiple comparisons, and statistical significance was set at *p* < 0.05; that is, the FDR was less than 0.05. The asterisks represent the statistical *p* value (* *p* < 0.05; ** *p* < 0.01; *** *p* < 0.001).

## 5. Conclusions

In conclusion, this work highlighted the potential importance of RAB family genes in the TME of HCC. Aberrant expression of RAB family genes is a nonnegligible factor in the TME heterogeneity and complexity of HCC. The models constructed based on RAB-associated signatures will contribute to improving our understanding of the characteristics of cell infiltration in the TME, and guide more effective immunotherapy strategies and prognostic assessments.

## Figures and Tables

**Figure 1 ijms-24-04335-f001:**
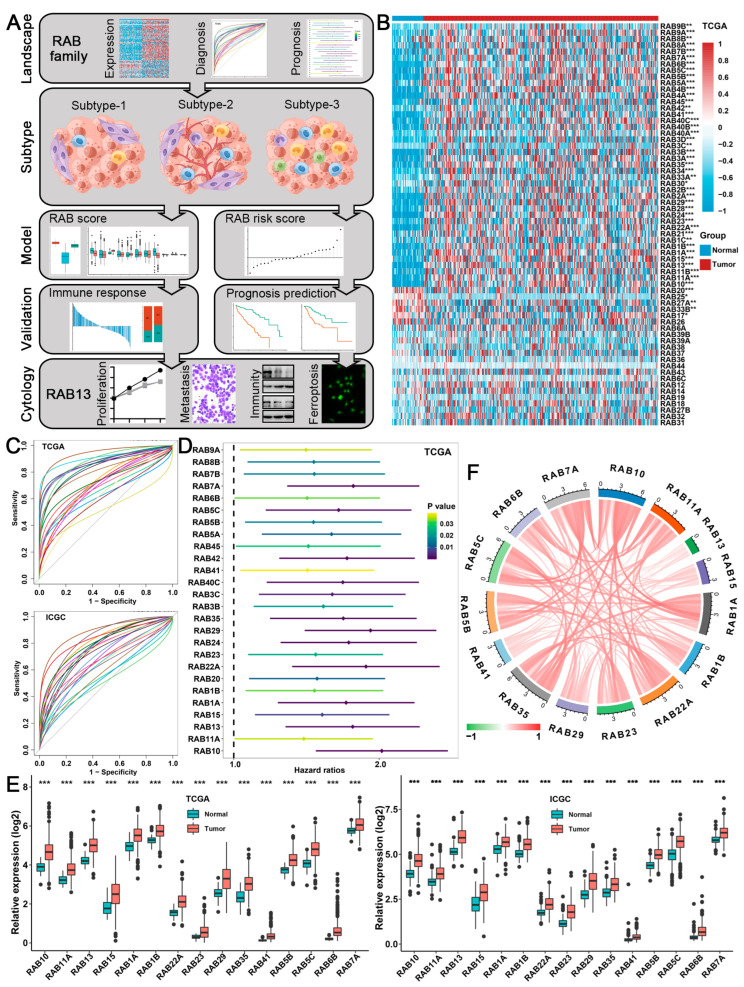
The expression landscape of RAB family genes in HCC. (**A**) Analytical workflow of this study. (**B**) Heatmap of the expression of RAB family genes in HCC and normal liver tissues from the TCGA cohort. (**C**) Receiver operating characteristic (ROC) curve of RAB family members among HCC patients in the TCGA and ICGC cohorts. (**D**) Hazard ratios of survival analyses for RAB family members in the TCGA cohort. (**E**) The relative expression of the 15 critical RAB family genes in the TCGA and ICGC cohorts. (**F**) The interaction among the 15 critical RAB family genes in HCC. The lines linking genes show their interactions, and the thickness of the lines shows the correlation strength. Positive correlations are marked with red lines, and negative correlations are marked with green lines. The asterisks represent the statistical *p* value (* *p* < 0.05, ** *p* < 0.01, and *** *p* < 0.001).

**Figure 2 ijms-24-04335-f002:**
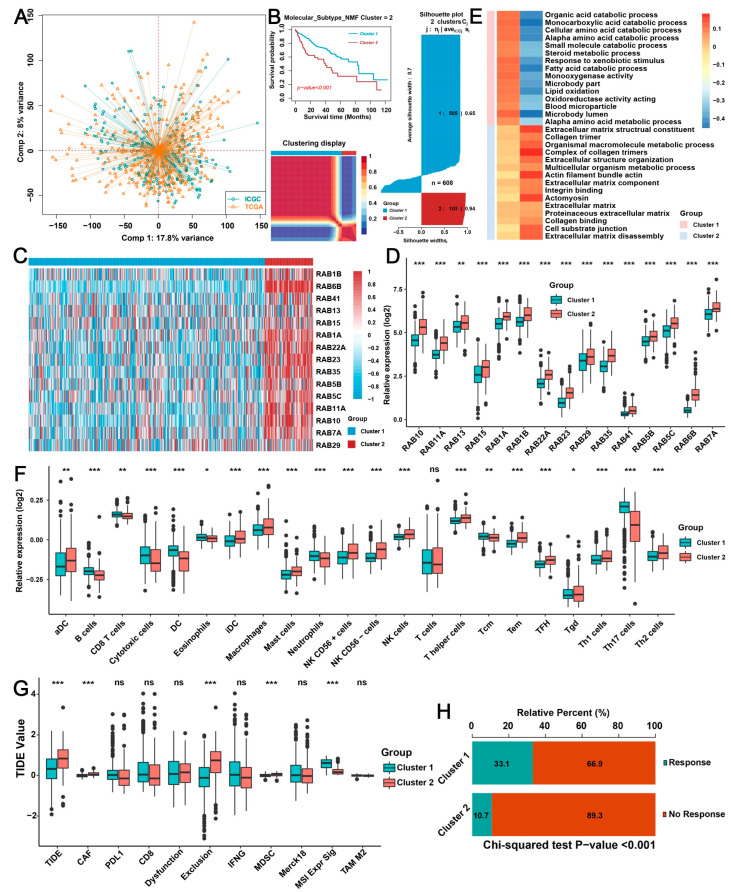
Clustering HCC based on the 15 RAB family genes by NMF analysis. (**A**) Principal component analysis (PCA) of the transcriptome profiles in the pooled HCC cohort from the TCGA and ICGC databases. (**B**) HCC samples were clustered by the NMF method. Kaplan–Meier survival curve for comparing the OS among two clusters by the “CancerSubtypes” package (upper panel). Two RAB clusters by PCA (lower panel). Silhouette width plots of NMF analysis (right panel). (**C**) Heatmap of the 15 critical RAB family genes in the 2 RAB clusters. (**D**) The relative expression of the 15 critical RAB family genes in the 2 RAB clusters from the pooled HCC cohort. (**E**) GSEA annotations for two RAB clusters in the pooled HCC cohort. (**F**) The diversity of immune cell infiltration patterns between patients with various RAB clusters is displayed. (**G**,**H**) TIDE values of two RAB clusters in the pooled HCC cohort. In (**H**), the chi-square test was used to calculate significant differences. The asterisks represent the statistical *p* value (* *p* < 0.05, ** *p* < 0.01, and *** *p* < 0.001). ns, no significance.

**Figure 3 ijms-24-04335-f003:**
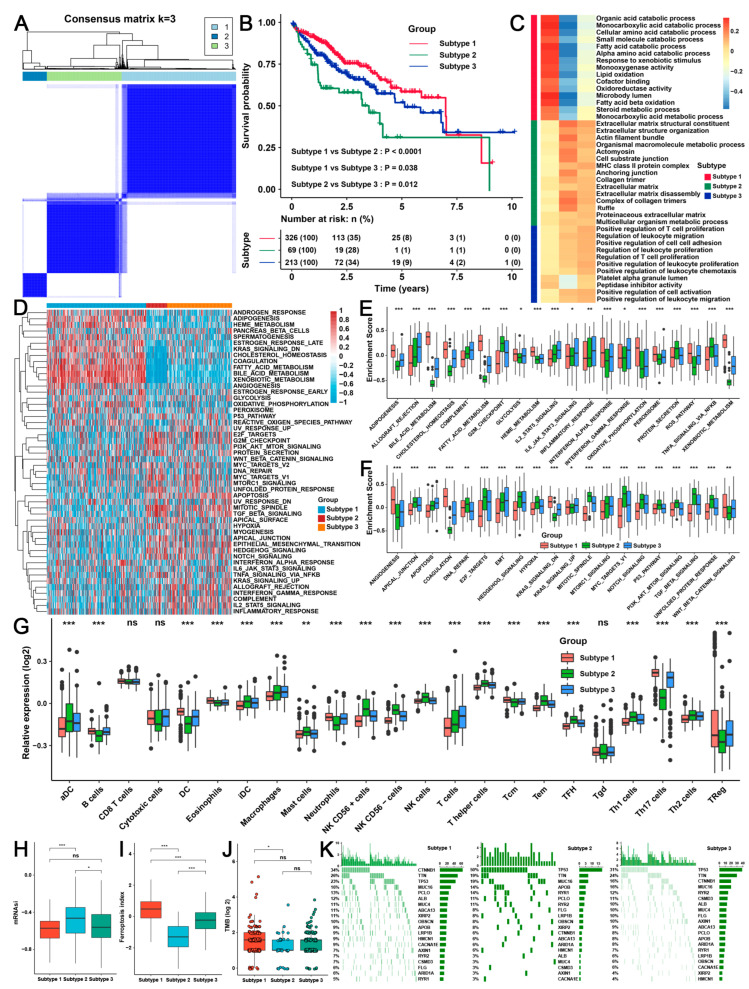
The three distinct RAB subtypes were identified in HCC using an unsupervised clustering analysis. (**A**) Consensus matrices of the pooled HCC cohort for k = 3. (**B**) Survival analyses for distinct RAB subtypes of the pooled HCC cohort. (**C**) The biological process from GSEA for distinct RAB subtypes in the pooled HCC cohort. (**D**) GSVA annotations for distinct RAB subtypes in the pooled HCC cohort. (**E**,**F**) GSVA annotations for distinct RAB subtypes in the pooled HCC cohort. (**G**) The diversity of immune cell infiltration patterns between patients with distinct RAB subtypes in the pooled HCC cohort is displayed. (**H**) mRNAsi score, (**I**) ferroptosis index, and (**J**) TMB score in distinct RAB subtypes of the pooled HCC cohort. (**K**) The gene mutation frequency in distinct RAB subtypes of the TCGA cohort. Each column represents an individual patient. The upper bar plot shows TMB. The number on the right indicates the mutation frequency in each gene. The right bar plot shows the proportion of each variant type. The asterisks represent the statistical *p* value (* *p* < 0.05, ** *p* < 0.01, and *** *p* < 0.001). ns, no significance.

**Figure 4 ijms-24-04335-f004:**
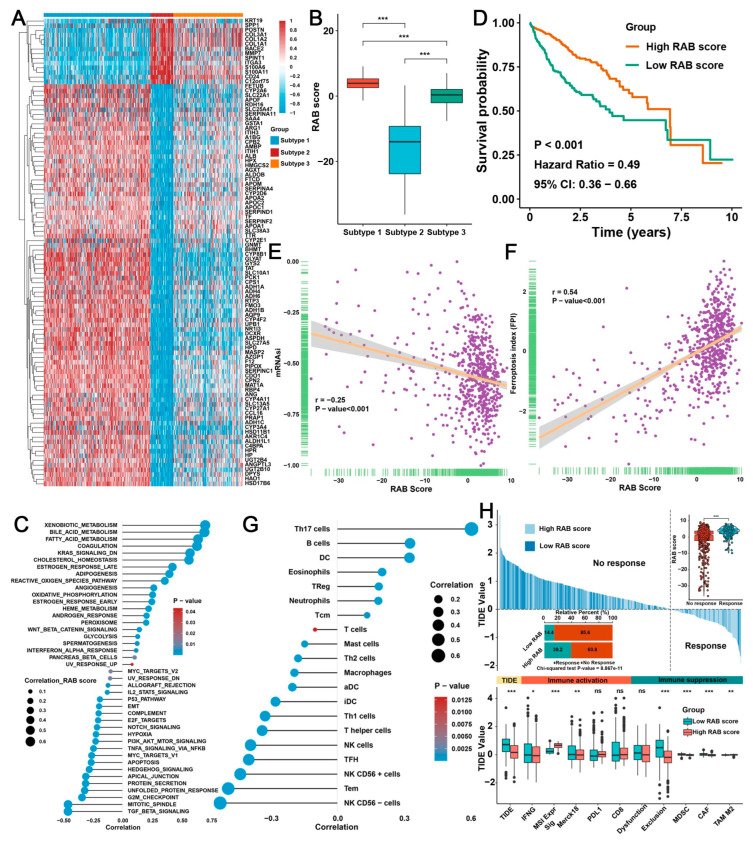
Construction of the RAB score for predicting immunotherapy response. (**A**) Heatmap exhibiting the expression of the 100 critical differentially expressed genes in the two RAB clusters. (**B**) The RAB score of distinct RAB subtypes in the pooled HCC cohort. (**C**) GSVA annotations show the correlation of the RAB score with the activation status of biological pathways. (**D**) Survival analyses for high- or low-RAB score groups of the pooled HCC cohort. (**E**,**F**) The correlation of RAB score with (**E**) mRNAsi score and (**F**) ferroptosis index in the pooled HCC cohort. (**G**) Correlation of the RAB score with the diversity of immune cell infiltration patterns in the pooled HCC cohort. (**H**) TIDE value of high- or low-RAB score groups of the pooled HCC cohort. The chi-square test was used to calculate significant differences. The asterisks represent the statistical *p* value (* *p* < 0.05, ** *p* < 0.01, and *** *p* < 0.001). ns, no significance.

**Figure 5 ijms-24-04335-f005:**
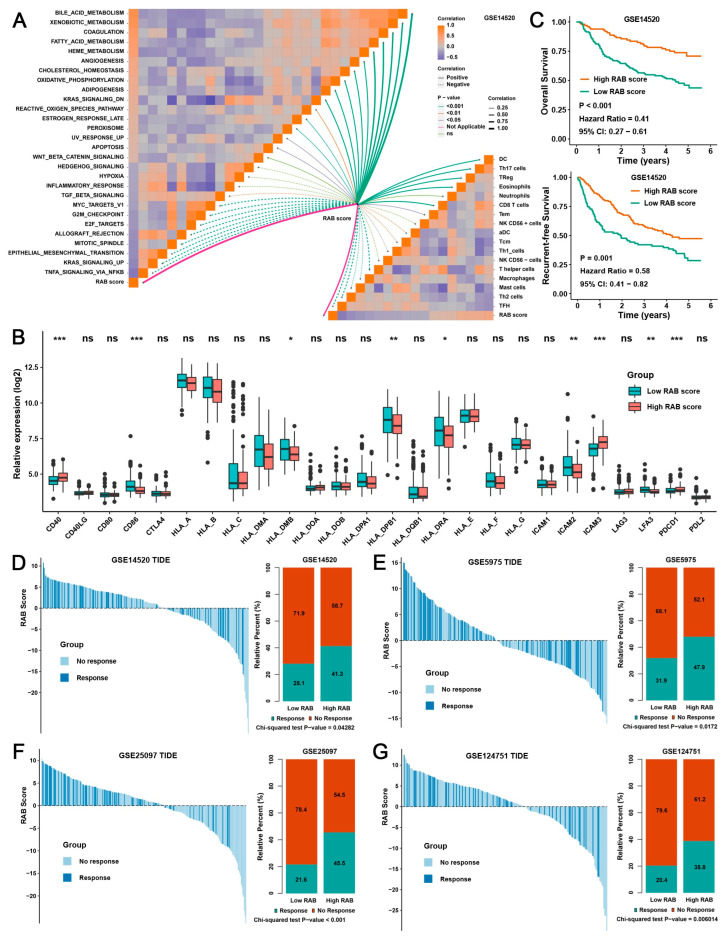
Validation of the RAB score for the immunotherapy response. (**A**) Correlation analysis of the RAB score with enriched signaling pathways from GSVA annotations (upper left panel) and previously published immune cell gene signatures (lower right panel) in the HCC cohort of GSE14520. (**B**) Differences in the expression of MHC molecules, costimulatory molecules, and adhesion molecules in distinct RAB subtypes of the HCC cohort from GSE14520. The upper and lower ends of the boxes represent the interquartile range of values. The lines in the boxes represent the median value, and the black dots show outliers. (**C**) Survival analyses for high- or low-RAB score groups of the HCC cohort of GSE14520. (**D**–**G**) TIDE values of the high- or low-RAB score groups of the HCC cohort from (**D**) GSE14520, (**E**) GSE5975, (**F**) GSE25097, and (**G**) GSE124751. The chi-square test was used to calculate significant differences. The asterisks represent the statistical *p* value (* *p* < 0.05, ** *p* < 0.01, and *** *p* < 0.001). ns, no significance.

**Figure 6 ijms-24-04335-f006:**
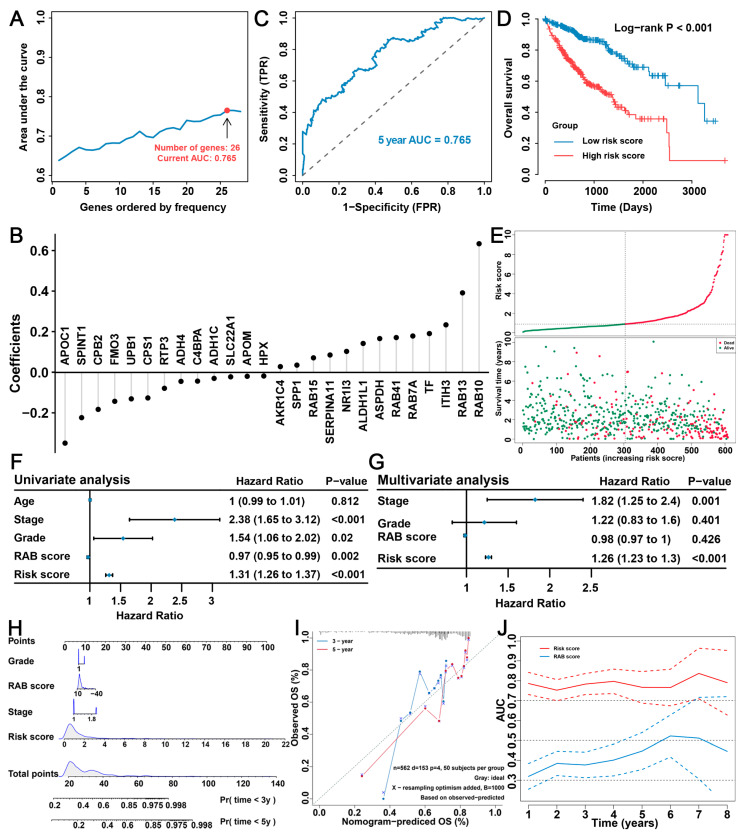
Construction of the RAB risk score for prognostic evaluation. (**A**,**B**) Iteration LASSO Cox regression analysis constructed a RAB risk score with 26 genes. (**C**) The AUC value was 0.765, and (**D**) the survival curve for a RAB risk score of the pooled HCC cohort is shown. (**E**) The survival time of each HCC patient with a different RAB risk score. (**F**) Univariate analysis and (**G**) multivariate analysis containing the RAB score, the RAB risk score, and clinical factors. (**H**) The comprehensive nomogram for predicting the probabilities of HCC patients with 3- and 5-year OS in the pooled HCC cohort. (**I**) The calibration plots for predicting HCC patients with 3- and 5-year OS in the pooled HCC cohort. The nomogram-predicted probability of survival is plotted on the *x*-axis; actual survival is plotted on the *y*-axis. (**J**) The time-dependent AUC values of the RAB score and the RAB risk score for the prediction of 3- and 5-year survival rates in the pooled HCC cohort.

**Figure 7 ijms-24-04335-f007:**
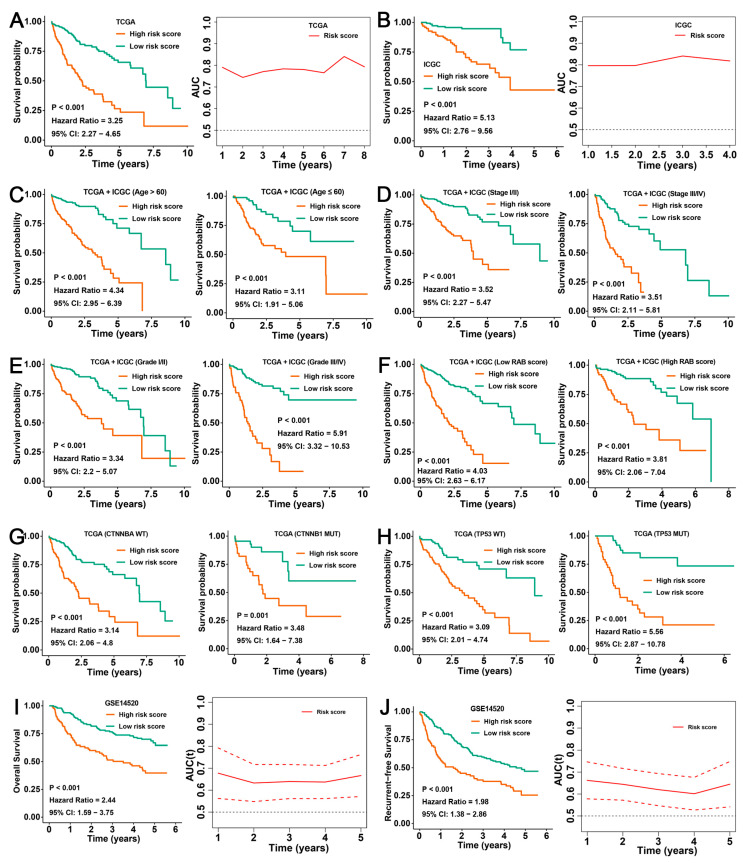
Validation of the RAB risk score for prognostic evaluation in HCC. (**A**,**B**) Kaplan–Meier curves of OS in patients with HCC from the (**A**) TCGA-LIHC and (**B**) ICGC-LIHC cohorts stratified by the RAB risk score. The time-dependent AUC values of the RAB risk score for the prediction of survival rates (right panel). (**C**) Kaplan–Meier curves of OS in HCC patients with different ages from the pooled HCC cohort stratified by the RAB risk score. (**D**) Kaplan–Meier curves of OS in HCC patients with different tumor stages from the pooled HCC cohort stratified by the RAB risk score. (**E**) Kaplan–Meier curves of OS in HCC patients with different histological grades from the pooled HCC cohort stratified by the RAB risk score. (**F**) Kaplan–Meier curves of OS in HCC patients with different RAB scores from the pooled HCC cohort stratified by the RAB risk score. (**G**,**H**) Kaplan–Meier curves of OS in HCC patients with different tumor mutation statuses stratified by the RAB risk score. (**I**,**J**) Kaplan–Meier curves of (**I**) OS and (**J**) RFS in patients with HCC from the GSE14520 cohort stratified by the RAB risk score. The time-dependent AUC values of the RAB risk score for the prediction of survival rates (right panel).

**Figure 8 ijms-24-04335-f008:**
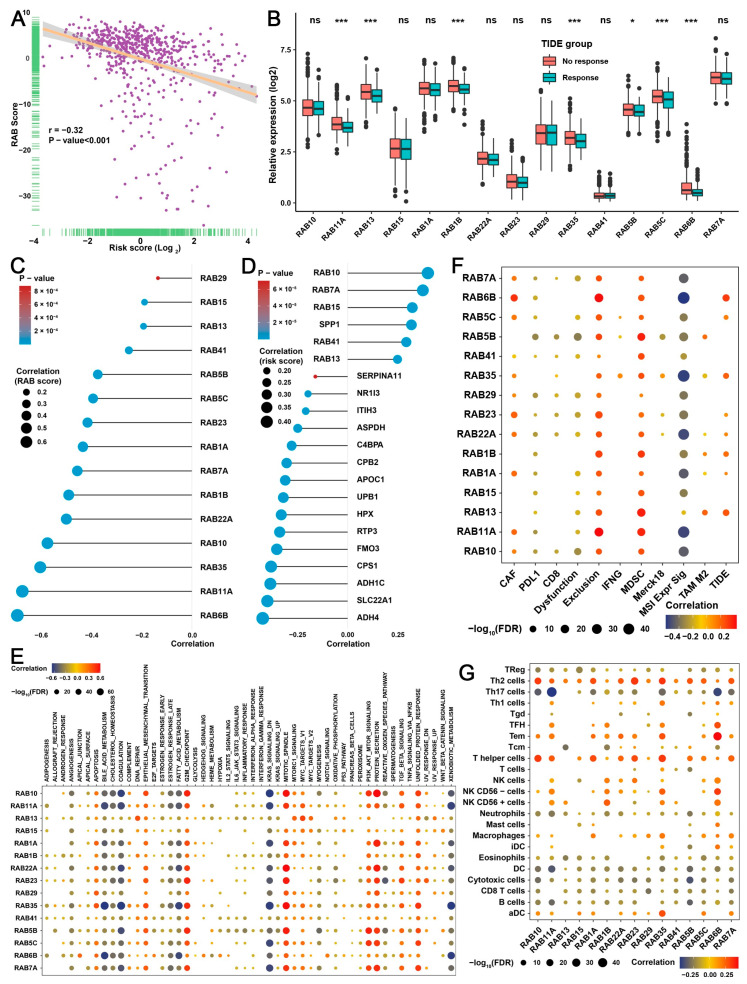
The integrative analysis identifies a potentially critical role for RAB13 in HCC. (**A**) Correlation between RAB score and RAB risk score in the pooled HCC cohort. (**B**) Relative TIDE values for high or low expression of the indicated RAB family genes from the pooled HCC cohort. The chi-square test was used to calculate significant differences. (**C**) Correlation of the RAB score with the indicated RAB family genes in the pooled HCC cohort. (**D**) Correlation of the RAB risk score with the indicated RAB family genes in the pooled HCC cohort. (**E**) Correlation of the indicated RAB family genes with enriched signaling pathways from GSVA annotations in the pooled HCC cohort. (**F**) Correlation of the indicated RAB family genes with immunosuppressive factors and immune cells from TIDE analysis in the pooled HCC cohort. (**G**) Correlation of the indicated RAB family genes with immune cell infiltration in the pooled HCC cohort. The asterisks represent the statistical *p* value (* *p* < 0.05 and *** *p* < 0.001). ns, no significance.

**Figure 9 ijms-24-04335-f009:**
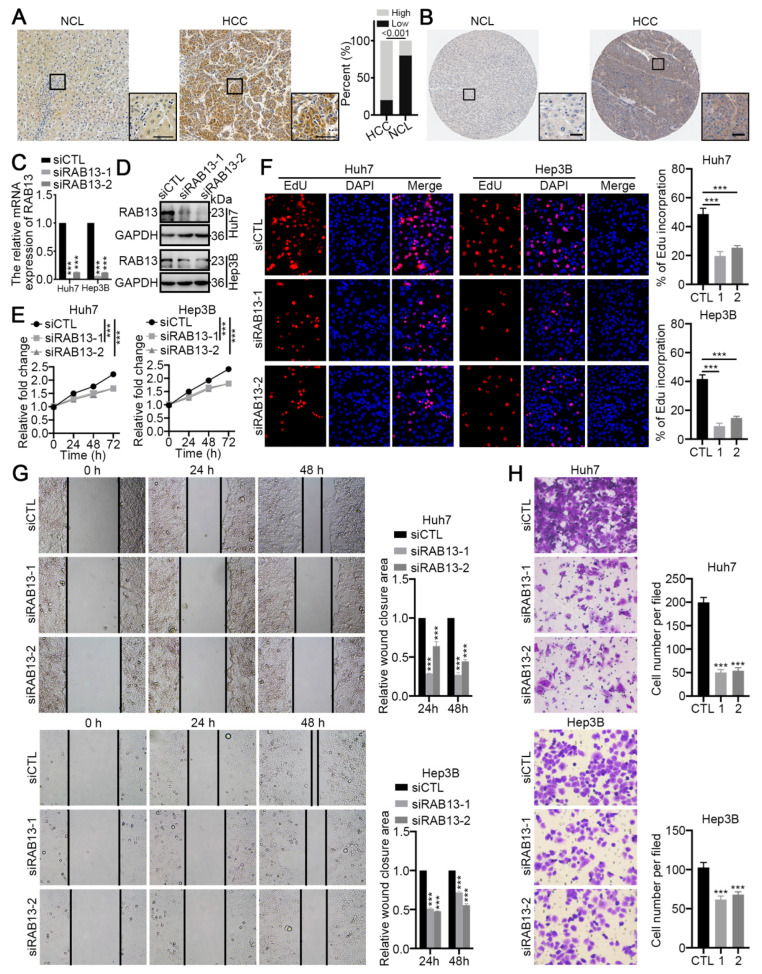
RAB13 promotes the proliferation and metastasis of HCC cells. (**A**,**B**) Representative IHC images of RAB13 in HCC and nontumor liver tissue (NCL) from (**A**) our clinical samples and (**B**) the Human Protein Atlas. The scale bar denotes 50 μm. (**C**) qRT–PCR and (**D**) Western blotting for RAB13 in Huh7 and Hep3B cells transfected with RAB13 siRNAs (siRAB13) and control siRNA (siCTL), respectively. (**E**) CCK-8 assays for Huh7 and Hep3B cells transfected with siRAB13 or siCTL over a 3-day period. (**F**) EdU assays for the indicated HCC cells transfected with siRAB13 or siCTL for 48 h. Magnification, 400×. (**G**) Wound healing (Magnification, 200×) and (**H**) Transwell assays for Huh7 and Hep3B cells transfected with siRAB13 or siCTL. Magnification, 400×. The results are presented as the means ± SDss, and three independent experiments (N = 3) were performed in triplicate. The student’s t-test was used for statistical analysis. The asterisks represent the statistical *p* value (*** *p* < 0.001).

**Figure 10 ijms-24-04335-f010:**
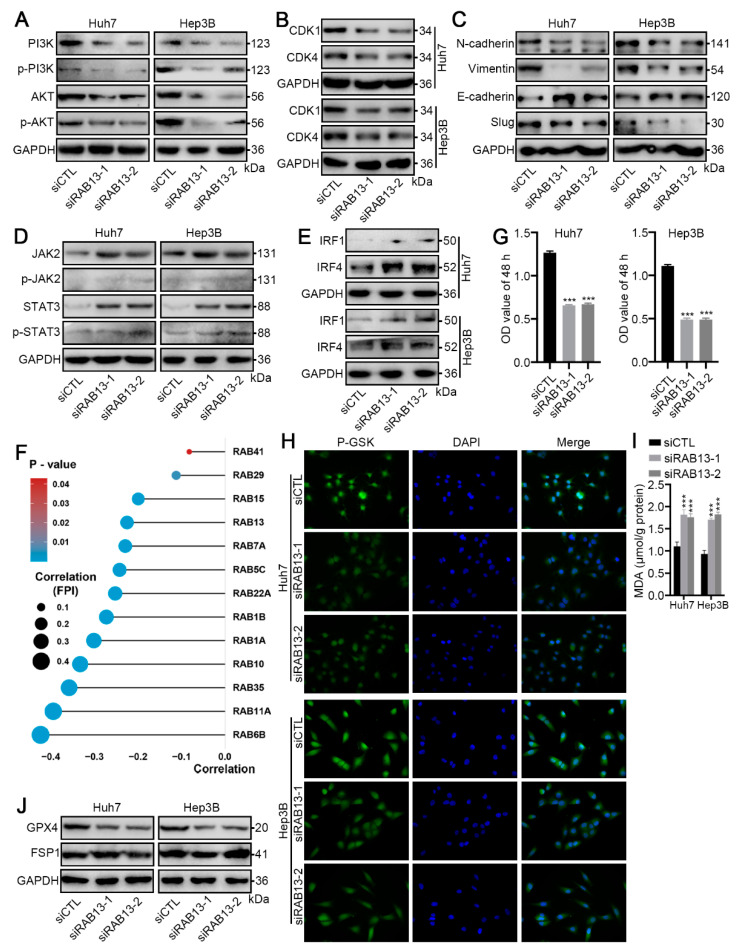
RAB13 knockdown promotes ferroptosis vulnerability. (**A**) Western blotting for PI3K/AKT signaling pathways in Huh7 and Hep3B cells transfected with siRAB13 or siCTL. (**B**) Western blotting for CDK1 and CDK4 in Huh7 and Hep3B cells transfected with siRAB13 or siCTL. (**C**) Western blotting for EMT signaling pathways in Huh7 and Hep3B cells transfected with siRAB13 or siCTL. (**D**) Western blotting for JAK2/STAT3 signaling pathways in Huh7 and Hep3B cells transfected with siRAB13 or siCTL. (**E**) Western blotting for IRF1 and IRF4 in Huh7 and Hep3B cells transfected with siRAB13 or siCTL. (**F**) Correlation of the indicated RAB family genes with ferroptosis vulnerability. (**G**) CCK-8 assays of Huh7 and Hep3B cells transfected with siRAB13 or siCTL and treated with 5 μM sorafenib for 72 h. (**H**) P-GSK staining of Huh7 and Hep3B cells transfected with siRAB13 or siCTL and treated with 5 μM sorafenib for 72 h. Magnification, 400×. (**I**) MDA levels in Huh7 and Hep3B cells transfected with siRAB13 or siCTL and treated with 5 μM sorafenib for 72 h. (**J**) Western blotting for GPX4 and FSP1 in Huh7 and Hep3B cells transfected with siRAB13 or siCTL. The results are presented as the means ± SDss, and three independent experiments (N = 3) were performed in triplicate. The student’s t-test was used for statistical analysis. The asterisks represent the statistical *p* value (*** *p* < 0.001).

## Data Availability

The datasets presented in this study can be found in online repositories. The names of the repositories/repositories and accession number(s) can be found in the article/Appendix A.

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
