# Peer review of "Construction of Two Independent RAB Family-Based Scoring Systems Based on Machine Learning Algorithms and Definition of RAB13 as a Novel Therapeutic Target for Hepatocellular Carcinoma"

_ijms, 2023, doi:10.3390/ijms24054335_

Round 1

Reviewer 1 Report (Previous Reviewer 1)

Thanks for the resubmitted manuscript, and the author also added an external validation of their own patients' cohort. I do not have any other suggestions for this work in the current format.

Author Response

Dear Editors and Reviewers,

Thank you for your letter and reviewers’ comments about our manuscript entitled “Construction of two independent RAB family-based scoring systems based on machine learning algorithms and definition of RAB13 as a novel therapeutic target for hepatocellular carcinoma”. Those comments are all valuable and very helpful for revising and improving our paper, as well as the important guiding significance to our research. We have studied the comments carefully and have made corrections point-to-point which we hope meet with approval. The responses to the reviewer’s comments are as follows:

  1. Thanks for the resubmitted manuscript, and the author also added an external validation of their own patients' cohort. I do not have any other suggestions for this work in the current format.

Response: Thanks for your suggestions. We appreciate the time and effort you dedicated to providing feedback on our manuscript and are grateful for the insightful comments and valuable improvements to our manuscript.

Reviewer 2 Report (New Reviewer)

In this study the Authors analyzed the expression landscape and prognostic significance of the RAB family in hepatocellular carcinoma and correlated these RAB genes family with tumor microenvironment features. The results contribute to generate models based on RAB-associated signatures will improve the knowledge of the characteristics of cell infiltration in the TME and guide more effective immunotherapy strategies and prognostic assessments.

Minor concerns:

1.      A better description of all acronym in the text should be made.

2.      The figures are very difficult to visualize.

3.      How was the immunohistochemistry cut-off defined? Has it already been used in the literature?

Author Response

Dear Editors and Reviewers,

Thank you for your letter and reviewers’ comments about our manuscript entitled “Construction of two independent RAB family-based scoring systems based on machine learning algorithms and definition of RAB13 as a novel therapeutic target for hepatocellular carcinoma”. Those comments are all valuable and very helpful for revising and improving our paper, as well as the important guiding significance to our research. We have studied the comments carefully and have made corrections point-to-point which we hope meet with approval. Revised portions are highlighted in the paper. The main corrections in the paper and the responses to the reviewer’s comments are as follows:

  1. A better description of all acronyms in the text should be made.

Response: Thanks for your suggestions. We have reviewed the acronyms in the manuscript and provided better explanations, and we have made corrections point-to-point which we hope meet with approval.

  1. The figures are very difficult to visualize.

Response: Thanks for your suggestions. We have carefully checked the manuscript figures, replaced Figure 2B, Figure 3H, Figure 3I, Figure 3J, Figure 3K, and Figure 4H with high-resolution original pictures, and adjusted the fonts accordingly for better visualization.

  1. How was the immunohistochemistry cut-off defined? Has it already been used in the literature?

Response: Thanks for your suggestions. RAB13 expression levels were classified as low if the score was less than 5 and high if the score was ≥ 5. Also, We have added the corresponding references (Line 652).

This manuscript is a resubmission of an earlier submission. The following is a list of the peer review reports and author responses from that submission.

Round 1

Reviewer 1 Report

In this study, the author aimed to reveal the role of the RAB family in hepatocellular carcinoma. They first develop three RAB subtypes and explore their relation with TME characteristics. Then, by using a machine learning algorithm, they established a RAB score to quantify TME features and immune responses of individual tumors, specifically in hepatocellular carcinoma. After the bioinformatics analysis, they further confirmed that the knockdown of RAB13, suppressed HCC cell proliferation and metastasis by inhibiting the PI3K/AKT signaling pathway, CDK1/CDK4 expression, and epithelial-mesenchymal transition. In addition, RAB13 inhibited the activation of JAK2/STAT3 signaling and the expression of IRF1/IRF4. More importantly, we confirmed that RAB13 knockdown enhanced GPX4-dependent ferroptosis vulnerability, highlighting RAB13 as a potential therapeutic target. However, there are some major points to improve.

1.      In the functional enrichment analysis, the author only used KEGG and GO-BP analysis. I suggest the author could also try hallmark gene sets, which are cancer-specific.

2.      In the analysis of the differentially expressed genes (DEGs) analysis, the author used a threshold as |log2FC| ≥ 1.0, I suggest the author use a higher cutoff like 2 or 2.5 to show a higher uniqueness of these DEGs genes.

3.      The author should show all the consensus matrices in the paper, not just when K=3.

4.      The results of RAB13 would be more credible if the author could do an external validation of their own patients' cohort (e.g. Survival analysis of RAB13 based on IHC).

5.      The bioinformatic analysis of RAB13 in Figure 8 was too complicated, I suggest the author could only focus on the role of RAB13 in hepatocellular carcinoma.

6.      There are some minor language errors. The authors should be revised the manuscript with an English language editor to make it more readable.

Author Response

Dear Editors and Reviewers,

Thank you for your letter and reviewers’ comments about our manuscript entitled “Construction of two independent RAB family-based scoring systems based on machine learning algorithms and definition of RAB13 as a novel therapeutic target for hepatocellular carcinoma”. Those comments are all valuable and very helpful for revising and improving our paper, as well as the important guiding significance to our research. We have studied the comments carefully and have made corrections point-to-point which we hope meet with approval. Revised portions are highlighted in the paper. The main corrections in the paper and the responses to the reviewer’s comments are as follows:

  1. In the functional enrichment analysis, the author only used KEGG and GO-BP analysis. I suggest the author could also try hallmark gene sets, which are cancer-specific.

Response: Thanks for your suggestions. Indeed, gene set variation analysis (GSVA) with hallmark gene sets was used widely in this study. We introduced the GSVA in the part of materials and methods, and some results of GSVA were shown in Figures 2E, 3E, 3F, and 4C.

  1. In the analysis of the differentially expressed genes (DEGs) analysis, the author used a threshold as |log2FC| ≥ 1.0, I suggest the author use a higher cutoff like 2 or 2.5 to show a higher uniqueness of these DEGs genes.

Response: Thanks for your suggestions. In fact, there are two screening conditions for DEGs. The first screening was conducted between RAB clusters, the purpose of this screening was to obtain as many DEGs as possible for subsequent analysis. So the criterion of this analysis was relative relax. The second screening was conducted between RAB subtypes with log2 |fold change (FC)| > 2, the purpose of this analysis was to narrow the DEGs for RAB score.

Our previous description was not accurate enough, we have revised the description of threshold setting in the part of materials and methods as follows on Page 23 Line 835:

“All statistical analyses were performed using R software (version 3.6.1). Analysis of differentially expressed genes (DEGs) between different defined groups was performed using the "limma" R package. DEGs between the two RAB clusters were obtained with significance criteria set as adjusted P value < 0.05 and log2 |fold change (FC)| > 1, while the criteria of P value < 0.05 and log2 |fold change (FC)| > 2 were set for screening DEGs between two RAB subtypes.”

  1. The author should show all the consensus matrices in the paper, not just when K=3.

Response: Thanks for your suggestions. We have added all the consensus matrices in the supplementary Figure S2D.

  1. The results of RAB13 would be more credible if the author could do an external validation of their own patients' cohort (e.g. Survival analysis of RAB13 based on IHC).

Response: Thanks for your suggestions. This will be an important step to be implemented for our next study of this molecule.

  1. The bioinformatic analysis of RAB13 in Figure 8 was too complicated, I suggest the author could only focus on the role of RAB13 in hepatocellular carcinoma.

Response: Thanks for your suggestions. The results presented in Figure 8 serve several purposes, 1) to fully elucidate and confirm the important role of the RAB family members. 2) to screen RAB13 as the next validated target. 3) to illustrate the potential role of RAB13 in HCC.

  1. There are some minor language errors. The authors should be revised the manuscript with an English language editor to make it more readable.

Response: Thanks for your suggestions. We have sent the article to AJE (https://www.aje.cn) for further language editing.

Reviewer 2 Report

In the current study the authors attempt to dissect the expression of RAB family of genes in hepatocellular carcinoma. They identify over-expression of these genes in the tumor, some of which also show association with survival. They further identify two clusters of HCC patients based on the expression of these genes and characterize the differences in survival, pathway and infiltrates in them. One of these clusters is predicted to respond favorably to immunotherapy. 

They then re-cluster the data based on differentially expressed genes from the previous cluster, though I'm not sure if that is the case, identifying 3 additional clusters and characterize there differences. From here they generate a RAB score which the authors claim predicts response to immunotherapy. They also generate RAB risk score using lasso cox regressions as a predictor of survival.  The authors then shortlist RAB13 based on its correlation with the RAB metrics, infiltration and pathway profiles which is then experimentally characterized using KO experiments as a gene required for HCC cell-line proliferation and metastasis. 

The some of these findings are interesting, the paper is poorly written, especially the methods often making it difficult to judge how and what the authors have done. Often some of the computational analysis and associated figures are convoluted. Further the figures legends and text do a poor job clarifying the analysis resulting in a fair bit of obfuscation. Several of the figure also report validation that was performed on the discovery datasets making them redundant. Over-all I think the paper require significant improvement in its writing, presentation, clarification of methods and removal of redundant figures and streamlining the analysis. As it stands I cannot recommend this paper for publication. I have listed my specific comments below

Major comments:

·         Please provide references for tools, datasets etc in the paper. Form what I can see Xena, Gtex, limma, GSVA, GSEA, KEGG Hallmarks, ICGC, TCGA among other aren’t referenced.   

·         The methods section indicates that the authors used FPKM values and limma for differential expression analysis. Which function in limma did the authors used? The use of FPKM for differential expression is discouraged. Also the limma function used for RNAseq is voom which takes counts as inputs and should not be provided with FPKM values? Without clear description of the precise analysis the authors perform it’s difficult to access the validity of their approach.

·         What was data from GTEx used for? It is mentioned in the methods and nowhere else? If GTEx was in fact used for how did the authors work around batch effects and differences in processing pipelines between GTEx, ICGA and TCGA.

·         For the pathway analysis please clarify what precisely was done.

o   Was DAVID used for over-representation analysis, if so how were the genes selected for the analysis? Was GO analysis also performed with DAVID? What were the statistical cutoff used to identify enriched pathways? None of these details are presented in the methods section or in the corresponding results and figure legends.

o    For the GSEA analysis what was the tool used for the analysis? Did the authors directly provide expression matrices or make use of ordered gene lists? What was the criteria used to identify differentially activated pathways?

o   GSVA  converts gene expression data to pathway activity using various approaches, which of these approaches did the authors utilize? Also consider using the same pathway definitions in the various analysis, jumping from one pathway definition to another lacks consistency and adds unnecessary confusion.

·         Please provide more details about what exactly TIDE analysis is? Also it’s not clearly why the TIDE plots include genes? How are the different from just gene expression values?

·         GSVA was used form estimating immune cell function, where did the signatures come from? Mentioning it came from an “authoritative study” is not a valid response. Further why didn’t the authors use abundance estimates from deconvolution approaches like cibersort? TCGA in fact has this data for most of its samples  (https://gdc.cancer.gov/about-data/publications/panimmune).

·         Many of the method sections (stemness, fere apoptosis index, transfection, pcr etc) simply reference papers without detailing the methods. In some cases these references are themselves secondary and reference other papers for their methods (reference 30 and 31). In other cases it’s unclear whether the authors used the output from the paper or generated it themselves. The stemness paper for instance (reference 34) provides multiple stemness measure for sample sin TCGA.

·         Figure 1A presents a pictographic workflow of the paper. Neither the text nor the figure legends provides any insight or details about this workflow. Baring the presentation of at least an outline describing the various steps I find this figure unnecessary.

·         Please consider replacing the adjusted P-value symbol with “q” or some other symbol instead of using “P” I kept confusing this with p-value. In fact it’s never quite clear to me when the authors are presenting p-value or p-values after correcting for multiple testing.

·         What is Figure 1C showing how is the AUC computed? What are the different color lines depicting? What was the criteria used to select RAB genes used in Figure 1D. Similarly in Figure 1E how was the selection performed, what cohort were the HR and AUC values thresholds applied in?

·         Page 4line 122: What is powerful hierarchical properties? In addition how would the authors know this prior to any analysis? Again the use of such qualitative informal language is problematic.

·         Page 5: why was NMF used here? NMF is often used for dimension reduction and the component loadings are then used for clustering. This is often useful in complex datasets with many dependent and independent factors. I’m not sure why NMF was used when just 15 features were used for the clustering and the same clustering can probably be easily captured simply through hierarchical clustering. It’s further perplexing as the authors used consensus clustering when performing clustering with a much large number of genes in Figure 3. The methods section for the NMF is also a mess, the text is more of a work salad rather than a coherent explanation of the method.  

·         The clustering in Figure 2 and 3 both show association with survival, however the authors don’t show if these clusters correlation with clinical valriables like age, stage, gender etc to dissociate the effect of RAB genes from them. Neither do they show that these clusters are independent form established molecular classifications. Which makes it difficult to judge if the signal is of genuine interest or just confounded by other well studied clinical and molecular variables.

·         Figure 3C is this GO analysis or results from GSEA? What exactly is plotted in the heatmap?

·         Figure 3D cannot be from GSEA. GSEA provides a single enrichment score as a summary statistics for two groups of samples being compared. GSVA on the other hand provides sample level activity scores. I also thing Figure 3E-F and D are likely the same. Again this underlines one of the major issues with this paper, it’s often unclear what form of pathway analysis was performed and how it was performed.

·         Figure 3G-K: The infiltration patterns doesn’t seem remarkably different from Figure 2. Also if this clustering was supposed to capture infiltration patterns why wasn’t the “immuno therapy response” analysis performed here? Also please explain how these data (mutation, infiltration etc) along with pathways activity patterns explain the survival data?

·         It’s also unclear how the RAB score itself was constructed. The methods section points to reference 17 which is not in English, however running I ran it through google translate and it doesn’t look like it has much to do with scoring RNAseq samples based on genesets.

·         Figure 4 B-H: uses TCGA and ICGC data to “validate” RAB score. As the score was generated from this datasets using genes that can differentiate groups of samples and reapplied to it this doesn’t count as validation.

·         Figure 5A is exceedingly convoluted, if the aim was to show correlation between RAB score with pathways why is pairwise correlation presented, most of these columns are irrelevant? Again what are GSEA annotations? As mentioned before GSEA produces summary statistics, I’m assuming the authors used single sample pathways activity from GSVA for their analysis?  

·         Why was the correlation and infiltration analysis restricted to a single samples (Figure 5A-B)?  

·         Figure 6 is largely illegible to me. The methods section is also scant on details making it difficult to evaluate. For instance how were the genes iterated over, what genes were used to build the model? Why are non RAB genes part of the model.

·         Figure 7 is unnecessary if a multivariate model was used to show that the risk score is an independent prognostic factor, it means it’s a prognostic factor even when the data is sliced and diced by other factors. Figure 7 is thus redundant and adds nothing in terms of interpretation of the result and just a whole lot of redundant statistics.   

·         Figure 8 uses a number of correlation analysis to justify RAB13 as a gene of interest for further analysis. Multiple other RAB’s in fact show much stronger trends in the criteria used, which makes the selection and focus n RAB13 rather arbitrary. There’s also little in the manuscript to suggest that the genes discarded are well understood and RAB13 isn’t.

·         The experimental data presented in Figure 9-10 deals with proliferation, metastasis and sensitivity to ferror apoptosis. None of these were particularly showcased in the 8 figures worth of computational analysis which seemed to indicate RABs play a role in inflammatory pathways, infiltration and immunotherapy response. I’m not quite sure how those observations segued to the experiments presented. These two figures are disjointed from the rest of the figure and to me don’t logically fit into this study. The authors need to provide a coherent connection between their computational analysis and experimental studies.    

Minor comments:

·         Page 2, line 51: I don’t quite understand how prominent is defined here? What makes RAB family the most prominent within the Ras super family? Using such informal descriptors without providing a coherent explanation as to why doesn’t seem appropriate.

·         The left panel for Figure 1E seems to be redundant with Fig 1B. Also what was the reason to look at the correlation between these genes? What do we infer from this and how does Figure  1F add to the development of the study from here?

·           Figure 2 C and D are essentially the same, the p-values can be reported as annotations in the heatmap, the boxplots are unnecessary.

·         Please clarify how genes for clustering in Fig 3A were selected? I suspect it’s done using differentially expressed genes comparing cluster in Fig 2, though it’s not entirely clear in the text.

·         Page 9 line 268: The same criteria picked up ~800 genes in the last section and only 100 genes in here? I’m also not entirely sure is clusters compare were clusters from Figure 2 or Figure3?

·         Figure 5 D-G please use different colors to indicate responders and non-responders, the current colors can be difficult to differentiate.  

·         Page 17 line 507 reference?

·         Page 17 line 517: How is this relevant? How do GPX4 and FSP1 regulate Ferro apoptosis. How do we interpret suppression of GPX4 in this context?

Author Response

Dear Editors and Reviewers,

Thank you for your letter and reviewers’ comments about our manuscript entitled “Construction of two independent RAB family-based scoring systems based on machine learning algorithms and definition of RAB13 as a novel therapeutic target for hepatocellular carcinoma”. Those comments are all valuable and very helpful for revising and improving our paper, as well as the important guiding significance to our research. We have studied the comments carefully and have made corrections point-to-point which we hope meet with approval. Revised portions are highlighted in the paper. The main corrections in the paper and the responses to the reviewer’s comments are as follows:

Major comments:

  • Please provide references for tools, datasets, etc in the paper. Form what I can see Xena, Gtex, limma, GSVA, GSEA, KEGG Hallmarks, ICGC, and TCGA among other aren’t referenced.

Response: Thanks for your suggestions. We have added relative references for tools and datasets (Xena, limma, GSVA, GSEA, KEGG Hallmarks, ICGC, TCGA).

  • The methods section indicates that the authors used FPKM values and limma for differential expression analysis. Which function in limma did the authors used? The use of FPKM for differential expression is discouraged. Also the limma function used for RNAseq is voom which takes counts as inputs and should not be provided with FPKM values? Without clear description of the precise analysis the authors perform it’s difficult to access the validity of their approach.

Response: Thanks for your suggestions. We noticed that the description of this part is inadequate in the part of materials and methods. In fact, we just downloaded FPKM values of LIHC in ICGC and TCGA, but we then transformed FPKM values into transcripts per kilobase million (TPM) values. We also know that the official recommendation is to use COUNT values for differential gene analysis, but given that we also used a lot of chip data, we finally chose the TPM value that is most comparable to the chip data as the first choice for analysis. The limma function voom was used to conduct data normalization and we have added a related description for this part as follows in Line 755 of Page 22: We then transformed FPKM values into transcripts per kilobase million (TPM) values.

  • What was data from GTEx used for? It is mentioned in the methods and nowhere else? If GTEx was in fact used for how did the authors work around batch effects and differences in processing pipelines between GTEx, ICGA and TCGA.

Response: Thanks for your suggestions. We also found this error, and this article does not cover the use of GTEx. We removed the description of this part, and this might be a version error caused by the revision of the previous submission.

  • For the pathway analysis please clarify what precisely was done.

Was DAVID used for over-representation analysis, if so how were the genes selected for the analysis? Was GO analysis also performed with DAVID? What were the statistical cutoff used to identify enriched pathways? None of these details are presented in the methods section or in the corresponding results and figure legends.

Response: Thanks for your suggestions. DAVID was just used to conduct KEGG pathways analysis for DEGs between two RAB clusters. GO analysis was not covered in this study and the GO analysis shown in Figure 3 legend was a writing mistake that had been fixed. The statistical cutoff used to identify enriched pathways was set as P-value < 0.05.

For the GSEA analysis what was the tool used for the analysis? Did the authors directly provide expression matrices or make use of ordered gene lists? What was the criteria used to identify differentially activated pathways?

Response: Thanks for your suggestions. We used “clusterProfiler” and “GSVA” packages in R to conduct GSEA analysis. We directly provide expression matrices and “c5.all.v6.2.symbols.gmt” file downloaded from the GESA database. We chose the top 15 pathways satisfied with P-value < 0.05 as differentially activated pathways.

GSVA converts gene expression data to pathway activity using various approaches, which of these approaches did the authors utilize? Also consider using the same pathway definitions in the various analysis, jumping from one pathway definition to another lacks consistency and adds unnecessary confusion.

Response: Thanks for your suggestions. We chose “gsva” method and “Gaussian” Kcdf as parameters of GSVA. And we have removed the duplicate genes involved in the hallmarks of gene sets. The genes used for the analysis all appeared independently in a particular set of pathways.

Please provide more details about what exactly TIDE analysis is? Also it’s not clearly why the TIDE plots include genes? How are the different from just gene expression values?

Response: Thanks for your suggestions. TIDE was the abbreviation of Tumor Immune Dysfunction and Exclusion. Based on tumor pre-treatment expression profiles, this TIDE module can estimate multiple published transcriptomic biomarkers to predict patient response to immunotherapy. The homepage of TIDE was http://tide.dfci.harvard.edu/. The output of TIDE contained several parts related to microenvironment conditions. The genes involved in TIDE might be normalized, so they represented the relative expression levels of these genes.

GSVA was used form estimating immune cell function, where did the signatures come from? Mentioning it came from an “authoritative study” is not a valid response. Further why didn’t the authors use abundance estimates from deconvolution approaches like cibersort? TCGA in fact has this data for most of its samples  (https://gdc.cancer.gov/about-data/publications/panimmune).

Response: Thanks for your suggestions. But due to network restrictions, we are unable to download the relevant data of panimmune data in TCGA. We also considered conducting cibersort, but the input data recommended FPKM values. We have many chip data and we have transformed the sequencing data to TPM values. So we chose GSVA to assess the immune cell infiltration levels. And the gene signatures were acquired from high standard studies.

Many of the method sections (stemness, fere apoptosis index, transfection, pcr etc) simply reference papers without detailing the methods. In some cases these references are themselves secondary and reference other papers for their methods (reference 30 and 31). In other cases it’s unclear whether the authors used the output from the paper or generated it themselves. The stemness paper for instance (reference 34) provides multiple stemness measure for sample sin TCGA.

Response: Thanks for your suggestions. Considering the length of the article, we cannot explain the specific operation process and steps of each analysis in detail in the article at one time. We believe that the references cited can be very helpful for readers to carry out relevant research, or at least we have done it through the study of these references. Readers are welcome to discuss these issues with us.

 Figure 1A presents a pictographic workflow of the paper. Neither the text nor the figure legends provides any insight or details about this workflow. Baring the presentation of at least an outline describing the various steps I find this figure unnecessary.

Response: Thanks for your suggestions. We consider this analytical workflow as a good way to show the idea and overview of the article, so as to facilitate readers' understanding.

  • Please consider replacing the adjusted P-value symbol with “q” or some other symbol instead of using “P” I kept confusing this with p-value. In fact it’s never quite clear to me when the authors are presenting p-value or p-values after correcting for multiple testing.

Response: Thanks for your suggestions. In fact, p-values were not corrected for multiple testing. There's no adjusted P-value here, it's all the original p-values.

What is Figure 1C showing how is the AUC computed? What are the different color lines depicting? What was the criteria used to select RAB genes used in Figure 1D. Similarly in Figure 1E how was the selection performed, what cohort were the HR and AUC values thresholds applied in?

Response: Thanks for your suggestions. Figure 1C showed the AUC of RAB family members in the TCGA and ICGC cohorts. Different color lines represented different RAB genes. HR was not shown in this analysis and the threshold of the AUC value was 0.7.

  • Page 4line 122: What is powerful hierarchical properties? In addition how would the authors know this prior to any analysis? Again the use of such qualitative informal language is problematic.

Response: Thank you for your comment and sorry for the misunderstanding. The descriptions here were obtained after we did a lot of analysis, and the relevant results in Figure 1 all illustrate this conclusion.

  • Page 5: why was NMF used here? NMF is often used for dimension reduction and the component loadings are then used for clustering. This is often useful in complex datasets with many dependent and independent factors. I’m not sure why NMF was used when just 15 features were used for the clustering and the same clustering can probably be easily captured simply through hierarchical clustering. It’s further perplexing as the authors used consensus clustering when performing clustering with a much large number of genes in Figure 3. The methods section for the NMF is also a mess, the text is more of a work salad rather than a coherent explanation of the method.

Response: Thanks for your suggestions. We used NMF to perform NMF on 15 important RAB genes in order to add survival variables so that a classification related to both survival and expression could be constructed. As for the use of consensus clustering in Figure 3, it is to analyze whether the expression of these genes is related, and whether there are differences in the survival of these gene subtypes is not the focus of our attention.

  • The clustering in Figure 2 and 3 both show association with survival, however the authors don’t show if these clusters correlation with clinical valriables like age, stage, gender etc to dissociate the effect of RAB genes from them. Neither do they show that these clusters are independent form established molecular classifications. Which makes it difficult to judge if the signal is of genuine interest or just confounded by other well studied clinical and molecular variables.

            Response: Thanks for your suggestions. The contents of Figure 2 and Figure 3 are only for a more comprehensive analysis of RAB genes, and the relationship between RAB gene expression patterns and survival is revealed by clustering. In the follow-up analysis, we gradually established the RAB score to predict survival. The above-mentioned stratified analysis was also analyzed in this part to illustrate the universality of the RAB score.

  • Figure 3C is this GO analysis or results from GSEA? What exactly is plotted in the heatmap?

Response: Thanks for your suggestions. Figure 3C was the result of GSEA, this describes an error and has been corrected.

  • Figure 3D cannot be from GSEA. GSEA provides a single enrichment score as a summary statistics for two groups of samples being compared. GSVA on the other hand provides sample level activity scores. I also thing Figure 3E-F and D are likely the same. Again this underlines one of the major issues with this paper, it’s often unclear what form of pathway analysis was performed and how it was performed.

Response: Thanks for your suggestions. Figure 3D was the results of GSVA, this describes an error and has been corrected.

  • Figure 3G-K: The infiltration patterns doesn’t seem remarkably different from Figure 2. Also if this clustering was supposed to capture infiltration patterns why wasn’t the “immuno therapy response” analysis performed here? Also please explain how these data (mutation, infiltration etc) along with pathways activity patterns explain the survival data?

Response: Thanks for your suggestions. The tumor microenvironment is complex and the results of a single analysis do not fully explain its association with survival data. We present the results of Figure 3G-K more to illustrate some of the differences in biological features between the three HCC subtypes we obtained.

  • It’s also unclear how the RAB score itself was constructed. The methods section points to reference 17 which is not in English, however running I ran it through google translate and it doesn’t look like it has much to do with scoring RNAseq samples based on genesets.

Response: Thanks for your suggestions. PCA analysis has been widely used for the clustering of molecular components, and we list references 17 for readers to better understand.

  • Figure 4 B-H: uses TCGA and ICGC data to “validate” RAB score. As the score was generated from this datasets using genes that can differentiate groups of samples and reapplied to it this doesn’t count as validation.

Response: Thanks for your suggestions. To increase the clinical sample size, we merged the TCGA and ICGC datasets using the algorithm. Subsequently, we used TCGA and ICGC respectively for the analysis of the RAB score which is a type of subset validation.

  • Figure 5A is exceedingly convoluted, if the aim was to show correlation between RAB score with pathways why is pairwise correlation presented, most of these columns are irrelevant? Again what are GSEA annotations? As mentioned before GSEA produces summary statistics, I’m assuming the authors used single sample pathways activity from GSVA for their analysis?

Response: Thanks for your suggestions. We have made a correction, GSVA not GSEA here for Figure 5A.

  • Why was the correlation and infiltration analysis restricted to a single samples (Figure 5A-B)?

Response: Thanks for your suggestions. Figures 5A-B are analyses for HCC with high/low RAB scores but are not restricted to a single sample.

  • Figure 6 is largely illegible to me. The methods section is also scant on details making it difficult to evaluate. For instance how were the genes iterated over, what genes were used to build the model? Why are non RAB genes part of the model.

Response: Thanks for your suggestions. We have illustrated this in the Figure legend. Here we further constructed the RAB risk score for prognostic assessment. Iterative LASSO Cox regression analysis was performed to construct a RAB risk score using 26 genes from the RAB score.

  • Figure 7 is unnecessary if a multivariate model was used to show that the risk score is an independent prognostic factor, it means it’s a prognostic factor even when the data is sliced and diced by other factors. Figure 7 is thus redundant and adds nothing in terms of interpretation of the result and just a whole lot of redundant statistics.

Response: Thanks for your suggestions. We present Figure 7 to fully illustrate the independent prognostic significance of our RAB risk score, both in terms of clinical parameters and mutation status.

  • Figure 8 uses a number of correlation analysis to justify RAB13 as a gene of interest for further analysis. Multiple other RAB’s in fact show much stronger trends in the criteria used, which makes the selection and focus n RAB13 rather arbitrary. There’s also little in the manuscript to suggest that the genes discarded are well understood and RAB13 isn’t.

Response: Thanks for your suggestions. We have stated in the manuscript that RAB13 was chosen as the gene for further experimental validation as RAB13 has not been studied in HCC. The comprehensive presentation of the 15 RAB family genes is intended to provide the reader with a more comprehensive understanding of the RAB family.

  • The experimental data presented in Figure 9-10 deals with proliferation, metastasis and sensitivity to ferror apoptosis. None of these were particularly showcased in the 8 figures worth of computational analysis which seemed to indicate RABs play a role in inflammatory pathways, infiltration and immunotherapy response. I’m not quite sure how those observations segued to the experiments presented. These two figures are disjointed from the rest of the figure and to me don’t logically fit into this study. The authors need to provide a coherent connection between their computational analysis and experimental studies.

Response: Thanks for your suggestions. We carried out the relevant descriptions before conducting these experiments. Notably, dysregulation of metabolic signaling regulates ferroptosis vulnerability. Therefore, we wondered whether RAB13 expression could alter the ferroptosis vulnerability of HCC cells. Besides, based on previous analysis, we further investigated the precise relationship of RAB13 expression with the PI3K/AKT signaling pathway, cell cycle regulation, and EMT.

Minor comments:

  • Page 2, line 51: I don’t quite understand how prominent is defined here? What makes RAB family the most prominent within the Ras super family? Using such informal descriptors without providing a coherent explanation as to why doesn’t seem appropriate.

Response: Thanks for your suggestions. Sorry for the misunderstanding, we have changed the “prominent” to “major” family member.

  • The left panel for Figure 1E seems to be redundant with Fig 1B. Also what was the reason to look at the correlation between these genes? What do we infer from this and how does Figure 1F add to the development of the study from here?

Response: Thanks for your suggestions. These results are only intended to provide a more complete picture of the expression characteristics of the 15 RAB family members.

  • Figure 2 C and D are essentially the same, the p-values can be reported as annotations in the heatmap, the boxplots are unnecessary.

Response: Thanks for your suggestions. We present these two figures to give patients a visual representation of the relative qualitative expression levels (Figure 2C) and the relative quantitative expression results (Figure 2D), respectively.

  • Please clarify how genes for clustering in Fig 3A were selected? I suspect it’s done using differentially expressed genes comparing cluster in Fig 2, though it’s not entirely clear in the text.

Response: Thanks for your suggestions. We have illustrated this in the Figure legend.

  • Page 9 line 268: The same criteria picked up ~800 genes in the last section and only 100 genes in here? I’m also not entirely sure is clusters compare were clusters from Figure 2 or Figure3?

Response: Thanks for your suggestions. We have illustrated this in the manuscript. According to the criteria of log |FC|>1 and P<0.05, a total of 830 DEGs were obtained. According to the criteria of log |FC|>2 and P<0.05, we further screened 100 DEGs.

  • Figure 5 D-G please use different colors to indicate responders and non-responders, the current colors can be difficult to differentiate.

Response: Thanks for your suggestions. This is the color defined by the author of the first proposed TIDE algorithm (PMID: 30127393) and we recommend using this color scheme to present the results.

  • Page 17 line 507 reference?

Response: Thanks for your suggestions. We have added references in Page 17 line 587 reference (Original page 17 line 507 reference).

  • Page 17 line 517: How is this relevant? How do GPX4 and FSP1 regulate Ferro apoptosis. How do we interpret suppression of GPX4 in this context?

Response: Thanks for your suggestions. We have illustrated this in the manuscript as follows: Finally, we detected alterations in GPX4 and FSP1 expression, which are two key targets that independently inhibit ferroptosis.

Round 2

Reviewer 1 Report

Validation of the authors own patients' cohort still need to be done.